

**Sensitivity of the WRF-Chem v4.4 ozone, formaldehyde, and their precursors simulations to multiple bottom-up emission inventories over East Asia during the KORUS-AQ 2016 field campaign**

Kyoung-Min Kim[1], Si-Wan Kim[2*], Seunghwan Seo[1], Donald R. Blake[3], Seogju Cho[4], James H. Crawford[5], Louisa K. Emmons[6], Alan Fried[7], Jay R. Herman[8,9], Jinkyu Hong[1], Jinsang Jung[10] Gabriele G. Pfister[6], Andrew J. Weinheimer[6], Jung-Hun Woo[11], and Qiang Zhang[12]

[1]Department of Atmospheric Sciences, Yonsei University, Seoul, South Korea
[2]Irreversible Climate Change Research Center, Yonsei University, Seoul, South Korea
[3]Department of Chemistry, University of California at Irvine, Irvine, CA, US
[4]Seoul Metropolitan Government Research Institute of Public Health and Environment, Gyeonggi-do, South Korea
[5]NASA Langley Research Center, Hampton, VA, US
[6]National Center for Atmospheric Research, Boulder, CO, US
[7]Institute of Arctic and Alpine Research, University of Colorado, Boulder, CO, US
[8]NASA Goddard Space Flight Center, Greenbelt, MD, US
[9]University of Maryland Baltimore County, Baltimore, MD, USA
[10]Korea Research Institute of Standards and Science, Daejeon, South Korea
[11]Department of Advanced Technology Fusion, Konkuk University, Seoul, South Korea
[12]Department of Earth System Science, Tsinghua University, Beijing, China

*To whom correspondence should be addressed. E-mail: siwan.kim@yonsei.ac.kr

Date: 01/12/2024

**Abstract**
In this study, the WRF-Chem v4.4 model was utilized to evaluate the sensitivity of $O_3$
simulations with three bottom-up emission inventories (EDGAR-HTAP v2, v3, and
KORUS v5) using surface and aircraft data in East Asia during the Korea-United States
Air Quality (KORUS-AQ) campaign period in 2016. All emission inventories were
found to reproduce the diurnal variations of $O_3$ and its main precursor $NO_2$ as compared
to the surface monitor data. However, the spatial distributions of the daily maximum 8-
hour average (MDA8) $O_3$ in the model do not completely align with the observations.
The model MDA8 $O_3$ had a negative (positive) bias north (south) of 30°N over China.
All simulations underestimated the observed CO by 50-60% over China and South
Korea. In the Seoul Metropolitan Area (SMA), EDGAR-HTAP v2, v3, and KORUS v5
simulated the vertical shapes and diurnal patterns of $O_3$ and other precursors effectively,
but the model underestimated the observed $O_3$, CO and HCHO concentrations. Notably,
the model aromatic VOCs were significantly underestimated with the three bottom-up
emission inventories, although the KORUS v5 shows improvements. The model
isoprene estimations had a positive bias relative to the observations, suggesting that the
Model of Emissions of Gases and Aerosols from Nature (MEGAN) version 2.04
overestimated isoprene emissions. Additional model simulations were conducted by
doubling CO and VOC emissions over China and South Korea to investigate the causes
of the model $O_3$ biases and the effects of the long-range transport on the $O_3$ over South
Korea. The doubled CO and VOC emission simulations improved the model $O_3$
simulations for the local emission dominant case, but led to the model $O_3$
overestimations for the transport dominant case, which emphasizes the need for
accurate representations of the local VOC emissions over South Korea.

## 1. Introduction

Air pollutants not only harm human health but also affect radiative balance, resulting in climate change (Anenberg et al., 2018; Franklin et al., 2015; Lee et al., 2014; Manning and von Tiedemann, 1995; Rosenzweig et al., 2008; Wild et al., 2001). Anthropogenic activities are the primary source of air pollutant emissions, which have significant temporal and spatial variability. Chemical transport models (CTMs) use bottom-up emission data to simulate ambient concentrations of air pollutants. CTMs then process these emissions, tracking their impact through chemistry, transport, and loss through deposition (Zhong et al., 2016). Therefore, sensitivity evaluations of CTMs to anthropogenic emission data are an essential part of atmospheric modeling research.

Several bottom-up emission inventories are available for chemical modeling of Asia, including the Multi-resolution Emission Inventory for China (MEIC), Regional Emission inventory in Asia (REAS), and Emissions Database for Global Atmospheric Research-Hemispheric Transport of Air Pollution (EDGAR-HTAP). Since 2010, Tsinghua University has developed the high-resolution MEIC emission inventory for China and updated the data to the v1.3, providing anthropogenic emissions by sector and species from 2008 to 2017 (Zheng et al., 2018). REAS provides emission data in Asia from 1950 to 2015 (Kurokawa and Ohara, 2020). In Europe, EDGAR-HTAP has been developed and widely used for CTM simulations from global to regional scale (Kim et al., 2021; Sharma et al., 2017; Sicard et al., 2021). Recently, EDGAR-HTAP v3 has been published, covering 19 years from 2000 to 2018 compared to only two years (2008 and 2010) in the version 2 data (Crippa et al., 2023). Zhong et al. (2016) compared REAS with EDGAR in July, 2007 over China, while Saikawa et al. (2017) compared 5 emission inventories including REAS, EDGAR, MEIC in China, without

validation. As bottom-up emission inventories are continuously updated for recent years,
there is an ongoing need to evaluate new emissions data.

3         The Ministry of Environment (MOE) in South Korea and National Aeronautics and

Space Administration (NASA) in the U.S. conducted the Korea-United States Air
Quality (KORUS-AQ) campaign in May-June 2016. The campaign provided a variety
of data sets, including ground-based and airborne observations, useful for the validation
of model simulations. The KORUS emissions, developed by Konkuk University, were
used by many modeling teams to simulate the air pollutant concentrations during the
campaign period. Numerous modeling studies were conducted for this period including
validations of CTM results with diverse observation datasets. Miyazaki et al. (2019)
adjusted emission inventories using various satellite data sets and Model for
Interdisciplinary Research on Climate with chemistry (MIROC-Chem), resulting
improved simulations of tropospheric $O_3$. Goldberg et al. (2019) reported
underestimations of NOx emissions in South Korea, particularly in Seoul. Souri et al.
(2020) also revealed the same issue in South Korea and conducted analysis of the
sensitivity of $O_3$ formation to adjustments in NOx and volatile organic compound (VOC)
emission derived from inverse modeling. Tang et al. (2019) revealed negative biases of
simulated CO concentrations in East Asia by utilizing satellite data and the Community
Atmosphere Model with Chemistry (CAM-Chem). Choi et al. (2022) modified
anthropogenic VOC emissions through the inverse modeling using satellite HCHO
observations with the Goddard Earth Observing System with Chemistry (GEOS-Chem),
which reduced $O_3$ and HCHO biases.

23         Recently, the updated version of bottom-up emission inventories and CTMs have

become available for the air pollution modeling studies in East Asia. In this study, we

selected the EDGAR-HTAP versions 2 and 3, and KORUS version 5 emission data and used the Weather Research and Forecasting model coupled with Chemistry (WRF-Chem) version 4.4 for intercomparison of the three emissions data sets, aiming to understand the status of precursor emissions from bottom-up emission inventories and their uncertainties, which may impact the $O_3$ formations in the model. $O_3$ and its major precursors were selected for model evaluation and the model results were evaluated with surface observation data in China and South Korea and aircraft data acquired over the South Korean peninsula and surrounding waters.

The manuscript is organized as follows. The data and methods section introduces emission inventories, the numerical model, and meteorological and chemical observations. The results section evaluates the model's meteorology and chemistry using routine surface observations over China and South Korea. Subsequently, the model results employing three bottom-up emission inventories are compared with sophisticated chemical observations obtained during the KORUS-AQ field campaign, primarily over South Korea. This comparison summarized the model's performance with each emission inventory. In the discussion section, strategies to enhance surface $O_3$ simulations, along with accurate precursor simulations, are proposed based on various emission scenarios for urban and regional areas over China and South Korea. The summary and conclusion section follow, providing overview of the key findings and conclusions drawn from the study.

## 2. Data and Methods

### 2.1. WRF-Chem model configurations

In this study, we utilized the WRF-Chem v4.4, which was developed by the National Oceanic and Atmospheric Administration (NOAA) and National Center for Atmospheric Research (NCAR), to simulate meteorological variables and chemical species in the atmosphere (Grell et al., 2005). The WRF-Chem v4.4 includes $N_2O_5$ heterogeneous chemistry that consists of several chemical reactions related with $ClNO_2$ and $N_2O_5$ reactions, resulting in nitrate aerosol. The reactions are incorporated in Secondary Organic Aerosol-Volatility Basis Set (SOA-VBS) with Regional Atmospheric Chemistry Mechanism (RACM) chemistry option (chem = 108) in WRF-Chem (Li et al., 2016).

We set 59 vertically customized eta (η) levels as vertical layers. The model's first layer height is approximately 40 m above ground level for the entire domain. The model's vertical layers are designed to include about 17 layers under 1.5 km to simulate planetary boundary layer chemistry and near surface vertical distribution in detail. The horizontal resolution is 28 x 28 $km^2$. The simulations in this study start at 12 UTC on April 24 and end at 12 UTC on June 11. The model meteorology restarts every 12 UTC (9 PM local time in South Korea) to minimize numerical errors. After the first 7 days of model initiation (spin-up), we analyzed the model results from May 1 to June 10. We used China standard time (+8 UTC) and Korea standard time (+9 UTC) for evaluations with observations. The model physics, chemistry, and aerosol schemes are summarized in Table S1 with corresponding references. The Global Forecast System (GFS) Final (FNL) analysis data are used for meteorological input and boundary conditions. The Community Atmosphere Model with Chemistry (CAM-Chem) output is used for

chemical boundary conditions (https://rda.ucar.edu/datasets/ds313.7/) (Buchholz et al.,

2019; Emmons et al., 2020). We used the Model of Emissions of Gases and Aerosols

from Nature (MEGAN) v2.04 to calculate biogenic emissions (Guenther et al., 2006).

We did not account for fire emissions because of small impact on air quality simulations

during the KORUS-AQ campaign period (Park et al., 2021). In our sensitivity

simulation with the Fire INventory from NCAR (FINN) v2.5 fire emissions

(Wiedinmyer et al., 2022), a marginal increase in the simulated averaged daily

maximum 8-hour average (MDA8) $O_3$ of approximately 1 ppbv (1.6 %) was noted in

China (Supporting information, Figure S1).

## 2.2. The model simulations using different anthropogenic emissions

### 2.2.1. Bottom-up emission data

EDGAR-HTAP v2, v3, and KORUS v5 anthropogenic bottom-up emission inventories

are compared with respect to their spatial distribution and total amount in Figure 1 and

Table S2. We applied the same diurnal factor for all three emissions data by species,

following the diurnal patterns for the Los Angeles Basin as in Kim et al. (2016) (also

see Figure S2).

EDGAR-HTAP v2 provides 2-dimensional emissions of $CH_4$, CO, $SO_2$, NOx (NO

+ $NO_2$), total non-methane volatile organic compound (NMVOC), $NH_3$, $PM_{10}$, $PM_{2.5}$,

BC, and OC in 2008 and 2010 with a horizontal resolution of 0.1˚ x 0.1˚. We used 2010

data since it is the most recent data available. The data are partitioned by each sector

and its sources such as air, ships, energy, industry, transport, residential, and agriculture

(https://edgar.jrc.ec.europa.eu/dataset_htap_v2). For East Asia, it included data from

the Model Inter-Comparison Study for Asia (MICS-Asia) and REAS v2.1. In South Korea, it adopted data from the Clean Air Policy Support System (CAPSS) (Janssens-Maenhout et al., 2015), and the underlying emission data had an original horizontal resolution of 0.25˚ x 0.25˚ over East Asia, which is resampled to 0.1˚ x 0.1˚ resolution by raster resampling and aggregation. The speciated EDGAR-HTAP v2 VOC data were obtained through the WRF-Chem site (https://www.acom.ucar.edu/wrf-chem/download.shtml) in the *anthro_emiss* program with the Model for Ozone and Related chemical Tracers (MOZART) species (Supporting Information, Table S3). The *anthro_emiss* program converts the EDGAR-HTAP v2 data into 28 x 28 $km^2$ grid by the RACM chemical species (Supporting Information, Table S4). It mapped the MOZART volatile organic compounds (VOC) species into the RACM VOC species (See the detailed equations in Supporting Information, Table S5) (Li et al., 2014; Emmons et al., 2010).

The EDGAR-HTAP v3 is extended to much longer time scale than the previous version EDGAR-HTAP (v2). The EDGAR-HTAP v3 covers 2000 to 2018 with a more detailed horizontal resolution (https://edgar.jrc.ec.europa.eu/dataset_htap_v3) (Crippa et al. 2023). While EDGAR-HTAP v2 uses MICS-Asia, only the REAS data are used in China and India in the EDGAR-HTAP v3. It adopts the CAPSS-Konkuk University (CAPSS-KU) data for South Korea and emission data provided by the Japanese government for Japan. We chose the data for 2016, according to the KORUS-AQ campaign period. Because the original EDGAR-HTAP v3 data provide VOC as total NMVOC with the unit of ton/month, we distributed the total NMVOC to MOZART VOC species with the ratio of each VOC species to total NMVOC from EDGAR-HTAP v2 in *anthro_emiss* program. Then, the assigned EDGAR-HTAP v3 data were again

converted to the RACM.

2         The KORUS v5 emission data represent 2016 in China and 2015 in other regions.

The Comprehensive Regional Emissions Inventory for Atmospheric Transport
Experiment (CREATE) v2.3 data from 2015 were used and the ship emissions from
CAPSS were added near the coastal region in South Korea (Jang et al., 2020; Woo et
al., 2012). The CREATE is originally developed by combining REAS, MEIC, Japan
Auto-Oil Program emission inventory (JATOP), and Korean Clear Air Policy Support
System (CAPSS). The NMVOC species from KORUS v5 were mapped following the
Statewide Air Pollution Research Center (SAPRC-99) mechanism, and we also
assigned the SAPRC-99 species to RACM (Carter, 2000) (Supporting information,
Table S5-6).

12         Figure 1 shows the spatial distribution of NO, CO, and TOL (toluene + less reactive

aromatics defined in RACM, see Table S4) emissions in May for each inventory. The
$NO_X$ emissions were assumed to be emitted as NO. The major cities in China and South
Korea had relatively high NO, CO, and TOL emissions, which are precursors affecting
$O_3$ formation. We define three boxes representing Eastern China, South Korea, and the
Seoul metropolitan area (SMA) and calculated the emissions (see Table S2). In South
Korea including SMA, EDGAR-HTAP v3 had the largest $NO_X$ emission among the
emission inventories. The KORUS v5 has lower $NO_X$ emissions in Eastern China by
46% and 39% compared to EDGAR-HTAP v2 and v3, respectively. The CO emission
was the lowest in EDGAR-HTAP v2 in South Korea, being 56% (69%) lower than that
in KORUS v5 (EDGAR-HTAP v3). KORUS v5 showed the highest CO emissions in
SMA though EDGAR-HTAP v3 showed more CO emissions in South Korea. However,
KORUS v5 had the smallest CO emissions in China, being 7% (9%) lower than that in

EDGAR-HTAP v2 (v3). The TOL emissions in KORUS v5 is higher than those in EDGAR-HTAP v2 (EDGAR-HTAP v3) by 176% (98%) in China. The relative difference of TOL between KORUS v5 and EDGAR-HTAP v2 (EDGAR-HTAP v3) is larger in South Korea by 263%. On the other hand, EDGAR-HTAP v3 has larger total NMVOC emissions over China than EDGAR-HTAP v2 and KORUS v5 by 38% and 27 %, respectively. These discrepancies in VOC emissions may lead to a change in the NOx/VOC-sensitive regime and $O_3$ production efficiency in South Korea and China. The sensitivity of $O_3$ formation to NOx emission has discrepancies by its regime; a reduction in NOx leads to a decrease in $O_3$ in the NOx-limited regime, while in the VOC-limited regime (or NOx-saturated regime), it results in an increase in $O_3$. This will be further discussed in section 3.2.

### 2.2.2. The model experiments

The model experiments are summarized in Table 1. The simulations using EDGAR-HTAP v2, v3, and KORUS v5 emissions are named as EDV2, EDV3, and KOV5, respectively. In this study, we found consistent underestimation of CO, HCHO, TOL, and XYL for all emissions by -40% ($\pm$ 2%), -25% ($\pm$ 1%), -67% ($\pm$ 21%), -53% ($\pm$ 18%), respectively, compared to observations from the DC-8 in South Korea. Here TOL and XYL are lumped species including toluene and xylene, respectively. This is in line with the results reported by Park et al. (2021), who found that almost every model underestimated CO. Underestimation of CO in East Asia is a well-known feature revealed by many studies. For example, Gaubert et al. (2020) mentioned that CAM-Chem underestimates CO during the KORUS-AQ campaign period and presented a CO

compensation method utilizing data assimilation with CO observations. Wada et al. (2012) pointed out that EDGAR v4.1 underestimates anthropogenic CO emissions in China by 45% compared to observation-based estimations of CO emissions. Moreover, underestimation of VOC is also found for all anthropogenic emission inventories. Kwon et al. (2021) estimated top-down emissions of anthropogenic VOCs utilizing Geostationary Trace gas and Aerosol Sensor Optimization spectrometer (GeoTASO). They found that top-down VOC emissions were up to 6.9 times higher than bottom-up emissions (KORUS v5).

For all emission inventories in simulations with WRF-Chem, $O_3$ is underestimated at most ground-based observation sites in South Korea. To figure out the potential causes of negative biases of $O_3$ in South Korea, we conducted three additional model simulations using EDGAR-HTAP v3 that shows the lowest bias of $O_3$ concentrations compared to DC-8 than EDGAR-HTAP v2 and KORUS v5 over the SMA; the mean biases are -16.9, -14.2, and -18.1 ppb with EDV2, EDV3, and KOV5, respectively. Two simulations are with twice the anthropogenic CO and VOC emissions in China (EDV3_Ch2) and South Korea (EDV3_Ko2), respectively, and the third simulation uses double CO and VOC emissions in both China and South Korea (EDV3_ChKo2) to investigate possible improvements in the simulated $O_3$ and CO from these emission changes. To simulate possible strategies to improve surface $O_3$ simulations over China and South Korea, we incorporated 4 additional emission scenarios involving the reduction of $NO_x$ and/or VOC emissions over China. Specifically, we considered the cases with a 50% reduction in $NO_x$ emissions only, a 50% reduction in VOC emissions only, a simultaneous 50% reduction in both $NO_x$ and VOC emissions, and a 75% reduction in $NO_x$ emissions only. For more details, refer to Section 4 (Discussion).

## 2.3. Observations

### 2.3.1. Meteorological data

The meteorological field that WRF-Chem reproduced is evaluated with the surface synoptic observation (SYNOP) data operated by the World Meteorological Organization (WMO) (http://www.meteomanz.com). Surface temperature, relative humidity, and surface wind speed are adopted for model validation. As the SYNOP data are provided every 3 or 6-hourly, we selected model data when the observation data are available. There were 271 sites in China-Taiwan-Hongkong and 48 sites in South Korea.

### 2.3.2. Ground-based observations

The surface observation network used in this study was obtained from Airkorea in South Korea and the China Ministry of Ecology and Environment (MEE) in China. The Airkorea observation network provides 1-hourly measurements of $NO_2$, $SO_2$, CO, $O_3$, $PM_{10}$, and $PM_{2.5}$ at suburban, background, roadside, city, and port sites (www.airkorea.or.kr). The concentrations of $NO_2$, CO, and $O_3$ are measured using the chemiluminescent, non-dispersive infrared, and ultraviolet photometric methods, respectively. In South Korea, there are indications of positive biases in $NO_2$ surface observations, potentially resulting in overestimations of ~30%, particularly at suburban sites in spring (Jung et al., 2017). The model data with 28 x 28 $km^2$ horizontal resolution were linearly interpolated to the 365 sites in South Korea, and we selected $NO_2$, $O_3$, and CO for model validation.

The Chinese observations were provided by MEE through the website (beijingair.sinaapp.com). Surface $NO_2$ over China was measured using a molybdenum converter, which has the potential for positive biases due to other $NO_2$-related oxidation products (Dunlea et al., 2007). CO was measured using infrared absorption (Zhang and Cao., 2015), and there were 1454 stations in China during the campaign period.

For validation of $NO_2$ and HCHO vertical column density, data from the Pandora spectrometer were used, which the model reproduced with emission inventories at the Olympic Park site (37.5232˚N, 127.126˚E). The HCHO data from Pandora is corrected because of internal off-gasing to avoid positive biases (Spinei et al., 2021). At the same observation site, surface $NO_2$ was also measured by a KENTEK NOx analyzer with photolytic method, and surface $O_3$ was measured using the same instrument. Ground-based HCHO was measured using Aerodyne QCL. We compared the observed diurnal cycle of vertical column and surface concentrations of $NO_2$ and HCHO with the model simulations utilizing EDV2, EDV3, and KOV5. We also used ground-based VOC data from gas chromatography flame ionization detector (GC-FID) operated by the Seoul Research Institute of Public Health and Environment (SIHE).

**2.3.3. Aircraft data**

The DC-8 research aircraft, operated by NASA, performed multiple flight measurements with a variety of measuring instruments. We utilized 1 minute interval merged data of $O_3$, $NO_2$, CO, HCHO, and VOC along the 20 flight paths (Figure 2). The nearest WRF-Chem grid is selected and then temporally and vertically interpolated to the aircraft data using linear interpolation to fully utilize the observations.

Atmospheric $NO_2$ and $O_3$ concentrations were measured using a 4-channel
chemiluminescence instrument, with an uncertainty of 100 pptv + 30% and 5 ppbv +
10%, respectively. CO concentrations were observed using a diode laser spectrometer,
with an uncertainty of 2% or 2 ppbv. The Compact Atmospheric Multi-species
Spectrometer (CAMS) was used to measure HCHO concentration, with a possible 3%
systematic error (Richter et al., 2015). We also utilized data from the Whole Air Sampler
(WAS) to analyze VOC species from different emission inventories (Colman et al.,
2001). In this study, we focused on DC-8 observations below a height of 2 km to
concentrate on planetary boundary layer (PBL) chemistry. The observation height was
determined by GPS altitude above ground level.

## 3. Results

### 3.1. The model meteorology simulations

The model temperature and relative humidity were compared with surface observations
in China and South Korea. The model-simulated temperature had a slight negative mean
bias of -0.91 ˚C (correlation coefficient R = 0.90) in China, with the largest negative
bias in southwestern China. In South Korea, the mean bias was -1.71 ˚C (R = 0.88). The
simulated relative humidity showed a negative bias of -20 to -10% in the North China
Plain (NCP) area and a positive bias of 10 to 20% in southwestern China. There was a
negative bias of relative humidity over the west coastal area and a positive bias of 10 to
20% at most observation stations in South Korea. The correlation coefficients between
the model relative humidity and observations were 0.85 and 0.76 for China and South
Korea, respectively. Overall, the comparisons showed decent model simulations of
meteorology. A negative temperature bias could result in a reduction of isoprene

emissions, as illustrated in Figure S3 of the Supporting Information, compared to the estimates based on accurately simulated temperature.

During the KORUS-AQ campaign period, WRF-Chem accurately simulated the daytime PBL height from a laser ceilometer (CL-31, Vaisala Inc., Finland) observed at Yonsei University in Seoul, South Korea (Lee et al., 2019). But, Travis et al. (2022) has indicated the possibility of PBL height underestimations by CTMs. Furthermore, due to limitations of the instrument, the ceilometer has potential to inadequately estimate nighttime PBL height. It is primarily attributed to the method based on aerosol gradients (Jordan et al., 2020). Therefore, the interpretation of simulated nighttime concentrations of air pollutants should be approached with caution. More analysis of meteorological fields, including PBL height, can be found in the Supporting Information (Table S7 and Figure S4-S5).

### 3.2. Evaluations with routine surface chemical observational data

The study compared simulated concentrations of $O_3$, $NO_2$, and CO with data from routine surface observational networks (Table 2 and Figure 3-7). First, the diurnal variations of the model $O_3$ using different emissions inventories were compared with observations for each subregion (Table 2 and Figure 3). Overall, all emission inventories successfully reproduced diurnal variations and absolute values of $O_3$ for most regions, but there were notable discrepancies in several regions.

In the North China Plain (NCP) region, EDV2 led to a negative model $O_3$ bias (-12 ppb) with R=0.65, while EDV3 and KOV5 simulated $O_3$ better with reduced biases and increased correlations (R=0.68-0.71). The high NOx emissions relative to the VOC

emissions in NCP led to a low formaldehyde to $NO_2$ ratio (FNR) (<1), suggesting that
the NCP area is in a VOC-limited regime with all emission inventories (Table 3). Due
to the elevated reactive VOC emissions in EDV3 and KOV5 compared to EDV2, both
EDV3 and KOV5 show improved $O_3$ simulations. Similarly, EDV2 had a negative $O_3$
bias (-17 ppb) with R=0.62 in the Yangtze River Delta (YRD) area, but EDV3 and
KOV5 much improved the simulations, which was also observed in the Northeastern
China (NEC) area. However, the model $O_3$ concentrations based on the three emission
inventories were overestimated in the Sichuan-Chongqing-Guizhou (SCG) and
Southeastern China (SEC) area. In SCG and SEC, the WRF-Chem simulated higher
biogenic isoprene emissions compared to anthropogenic TOL and XYL emissions by
up to a factor of 10, leading to a high FNR (> 1). In Pearl River Delta (PRD), EDV2
showed the lowest bias (-0.3 ppb) compared to EDV3 and KOV5 because EDV3 and
KOV5 have elevated anthropogenic VOC emissions as well as enhanced biogenic
isoprene emissions under a VOC-limited regime (Table 3). In the suburban area of
Northern China (NOC), all emission inventories reasonably simulated hourly $O_3$
concentrations.
Averaged $O_3$ was well simulated in South Korea (KOR) with low biases (-1 to 0.7
ppb), but a negative bias appears over the Seoul metropolitan area (SMA) with all
emissions (-5.5 to -3.5 ppb) (Table 2). WRF-Chem simulations indicate SMA as a
highly NOx-saturated region (FNR < 0.2), resulting in being VOC-sensitive for $O_3$
production. The underestimated model $O_3$ levels in this region suggests the possibility
of insufficient anthropogenic VOC emissions in SMA across all emission inventories
(Table 3). A detailed discussion will be provided in section 3.3.
The study also analyzed the mean values of MDA8 $O_3$ concentration at each site
and their spatial distributions for the entire campaign period (Figure 4). The spatial
distributions of the model MDA8 $O_3$ were not well correlated with those of the
observations. But, notable disparities were observed in simulating MDA8 $O_3$ when the
different emissions were used. For the north and eastern part of China including Beijing
and Shanghai, large negative biases disappear when using EDV3 and KOV5. KOV5
only shows a significant correlation with the surface MDA8 $O_3$ observations (including
929 sites) than EDV2 and EDV3 in China (0.43 versus 0.01, 0.20). The correlations
between the time series of the model MDA8 $O_3$ and observations varied at each site,
with about 40-60% of sites (depending on the emission inventories) showing a
correlation coefficient greater than 0.6 (see Supporting Information, Figure S6), and the
locations of these sites were scattered. The correlation slightly improved with hourly
$O_3$ concentrations instead of MDA8 $O_3$, with about 50-60% of sites having a correlation
coefficient greater than 0.6 (Supporting Information, Figure S6). For this metric, high
correlations occurred in pollution hot spots north of 30°N and the South Coast of China,
in which the ratio of HCHO to $NO_2$ (FNR) was much less than 1, suggesting VOC-
limited/NOx-saturated chemical regime (Supporting Information, Figure S7). The
model MDA8 $O_3$ were underestimated for the pollution hot spots with a low HCHO to
$NO_2$ ratio located north of 30°N, suggesting a possibility of model underestimations of
anthropogenic VOC emissions causing model MDA8 $O_3$ biases at these sites. In
contrast, the simulated MDA8 $O_3$ was generally overestimated for sites south of 30°N
in which HCHO concentrations were high (Supporting Information, Figure S7). Zhang
et al. (2020) reported that simulated biogenic isoprene (ISO) from MEGAN was
overestimated compared to observation sites south of 35˚N in China.
The EDV2 and EDV3 showed a positive $NO_2$ bias over the YRD, NCP, and PRD
regions, which include large cities in China (Table 2 and Figure 5-6). On the other hand,
EDV2 and EDV3 had small negative $NO_2$ biases in the NEC and NOC regions. All
models demonstrated reasonable $NO_2$ model performance in the SCG region, where
MDA8 $O_3$ was overestimated (Figure 3 and 5). In the YRD region, there were large
positive $NO_2$ biases with EDV2, EDV3, and KOV5 (ranging from 6.4 to 22.7 ppb). Liu
et al. (2021) reported that YRD is in a VOC-limited regime when using EDV2. The
findings indicated that a reduction in NOx emissions led to an increase in $O_3$
concentrations, while a reduction in VOC emissions resulted in lower $O_3$ concentrations.
The $O_3$ in YRD can be attributed to the combined influence of higher anthropogenic
NOx emissions and VOC originated from both anthropogenic and biogenic sources
(Figure S7). In contrast, KOV5 underestimated $NO_2$ in the NCP region, while EDV2
and EDV3 did not. All emissions showed significant discrepancies compared to $NO_2$
observations in the SEC area, with a low correlation coefficient (0.19 to 0.26). EDV3
showed the smallest bias of -1.9 ppb (-0.8 ppb) compared to EDV2 and KOV5 in South
Korea (SMA). The daily averaged $NO_2$ exhibited spatial distributions similar to MDA8
$O_3$ and CO (Figure 6). The slopes of regression between the three model simulations
and observations were 1.31, 1.03, and 0.8 for EDV2, EDV3, and KOV5, respectively,
in China. The correlation coefficients between the simulated $NO_2$ utilizing EDV2,
EDV3, and KOV5 and surface data were around 0.6 in China. EDV2, EDV3, and
KOV5 demonstrated good correlations with observations in South Korea (R = 0.69-
0.74). Correlation coefficient (R) was the highest with KOV5 in South Korea (R=0.74).
Likewise, the diurnal patterns of Ox (= $NO_2 + O_3$) are well simulated with all
emission inventories (Supporting Information, Figure S8). The spatial distribution and

diurnal patterns of Ox are similar to $O_3$ except YRD (Supporting Information, Figure S9). In YRD, the overestimations of Ox with all emission inventories reveals that same issue of $NO_2$ overestimations in Figure 5. Even though $O_3$ is well simulated in this region, the negative impact of NOx titration to $O_3$ formation is compensated with the overestimated anthropogenic and biogenic VOC emissions as mentioned above.

The simulated CO was averaged at each site and compared with observations during the KORUS-AQ campaign period (Figure 7). The three model results showed similar spatial distributions to observations, indicating higher CO concentrations in the NCP, YRD, and PRD regions than their surrounding areas. However, all simulations failed to reproduce the abundance of CO, indicating large negative biases throughout the country. The bias was larger in South Korea than in China.

## 3.3. Evaluations with the airborne and special surface chemical observations during KORUS-AQ

### 3.3.1. The aircraft observations

Figure 2 shows the flight paths flown by the DC-8 during the KORUS-AQ campaign period. In Table 4, we compare the model results for $O_3$, $NO_2$, CO, HCHO, TOL, XYL, ETE (Ethene or OL2), and ISO with the corresponding observed values for all flight tracks under 2 km height in South Korea. On average, the model underestimated $O_3$ by 15-18 ppb, with EDV3 exhibiting the lowest $O_3$ bias (-15.1 ppb) compared to EDV2 and KOV5 (-16.8 and -17.5 ppb, respectively). All emissions showed positive biases for $NO_2$ (0.64 to 1.72 ppb), ETE (0.08 to 0.14 ppb), and ISO (0.1 to 0.11 ppb). However, the model significantly underestimated CO, HCHO, TOL, and XYL for all three emissions. Given the large spatial variability of air pollutants in South Korea, we also

sampled aircraft data from six regions (see Figure 2) and compared the three model
results with the aircraft observations under 2 km height (Figure 8).

3        The flight tracks that surveyed large power plants and factories in the Chungnam

region on a daily basis are shown in Figure S10 in the Supporting Information. The
largest negative model $O_3$ bias was observed over the Chungnam region, with a
difference of 38-41 ppb. Emission estimation uncertainties can be significant over this
region, where there are large point sources such as coal-burning power plants and
petrochemical industries. The model $NO_2$ agreed with the aircraft observations in SMA,
but it tends to overestimate the measurements in the other areas. There were substantial
model overestimations of $NO_2$ with EDV3 over the Chungnam and Busan areas, while
KOV5 showed the most reasonable model $NO_2$ simulations. The model CO near the
surface was underestimated in the entire domain, resulting in high negative model CO
biases relative to the aircraft observations across the six regions (Figure 8). We
additionally conducted a sensitivity test to investigate the contribution of CO to $O_3$
concentrations in SMA (Supporting information, Figure S11). Doubling CO emissions
in China did not significantly change $O_3$ concentrations at all levels under 2 km. Only
1.4 ppb of $O_3$ concentrations are changed on average during all flight observations.

18       We also evaluated the model HCHO, which can be formed by oxidation of other

VOCs but also directly emitted by anthropogenic sources, to investigate uncertainties
in anthropogenic VOC emissions. The model HCHO was underestimated by all
emission inventories for all subregions, with negative biases being evident in the SMA,
Yellow Sea, and Chungnam regions.

23       Other model VOC species, such as TOL, XYL, ETE, and ISO, were also analyzed.

These VOC species are classified by their chemical structures and reactivities in the
RACM (Stockwell et al., 1997) (Table S4). For example, TOL includes toluene and
relatively less reactive aromatics, while XYL includes xylene and more reactive
aromatics. The WAS data from DC-8 were lumped into RACM (Supporting
Information Table S8, Lu et al., 2013) and were compared with aircraft observations.
When the model TOL or XYL was compared with the observed toluene and xylene, the
model using KOV5 reasonably reproduced the observed concentrations (light gray bars
in Figure 8). However, the model TOL (even using KOV5) underestimated the observed
lumped TOL for most of the regions except for Busan (bars including the dark gray part
in Figure 8). The model using KOV5 reasonably reproduced the observed xylene or
XYL, except for the Chungnam and Busan regions. The observed ethene (or ETE)
concentrations were low (< 0.5 ppb), except for the Chungnam region, where the
average of measurements was 2.1 ppb. The model ethene concentration was higher than
the observations for the SMA, Kyungbuk, and Busan regions, while it had a large
negative bias (-1.6 ~ -1.3 ppb) for the Chungnam region. Regarding ISO, one of the
most important biogenic VOCs, the model values were larger than the observations by
a factor of 2.

17       In summary, underestimated CO and aromatic VOCs are the main features of the

model evaluation with aircraft observations, along with underestimated ozone and
HCHO. The largest discrepancies occur over the Chungnam region, where large point
sources are located on the west coast of South Korea. The detailed statistics over the
SMA and Chungnam area can be obtained from the Supporting Information (Table S9-
S10).

23       Figure 9 displays the vertical distributions of observed and simulated $O_3$ and related

species over SMA. The shapes of the simulated profile were in agreement with the

observations. Particularly, the model accurately reproduced the observed $NO_2$ profiles though the surface $NO_2$ is underestimated by -4.2 to -0.8 ppb in SMA (Table 2 and Figure 9b). The underestimation of simulated surface $NO_2$ is explained by the overestimation of molybdenum converter method; surface concentrations of $NO_2$ from molybdenum converter is larger than photolytic converter by 13.6% on average and 64% at 4 pm (Figure 10). Although the diurnal pattern of surface $NO_2$ at 12-20 LT is explained by the overestimation by measurements using a molybdenum converter, there are still some other possible reasons; 1) the emission diurnal profile used in this study was developed for the Los Angeles Basin, which may need to be adjusted for SMA, 2) the uncertainty of HOx and ROx radicals from other sources can affect the $NO_2$ concentrations.

However, the simulated $O_3$ and HCHO had negative biases of 16.4 ppb and 0.73 ppb, respectively, persisting from the surface to 2 km. Additionally, the simulated CO underestimated the observations by 40% throughout the vertical layer. While the model TOL and XYL, utilizing KOV5, agreed well with the observations at surface level and had the lowest bias of -0.88 and -0.12 ppb under 2 km, the results using EDV2 and EDV3 substantially underestimated the observations throughout the layer (Supporting information, Table S9). On the other hand, the model-simulated ETE and ISO overestimated the observations below 1 km over SMA.

### 3.3.2. The ground-based observations

During the KORUS-AQ campaign, Pandora and surface measurements were co-located at Olympic Park. Figure 10 compares the observed diurnal cycle of Pandora vertical columns and surface concentrations of $NO_2$ and HCHO with the model simulations.

NO$_2$ measurements were made using a photolytic converter, providing better accuracy compared to measurement with a molybdenum converter. All emissions reasonably simulated the diurnal patterns of vertical column and surface NO$_2$ and HCHO concentrations.

The surface NO$_2$ peak appeared at 07 LT in the model and 08 LT in the observations, associated with the increase of traffic and the under-developed convective boundary layer. On the other hand, the Pandora NO$_2$ column amount increased from 06 LT to 12 LT and stayed at that value throughout the afternoon, indicating the increase of NO$_x$ emissions from morning to afternoon. The model-simulated NO$_2$ columns agreed with those from Pandora in terms of absolute values and diurnal variations. The opposite patterns between surface and column NO$_2$ were attributed to the change of boundary layer height; NO$_2$ is concentrated near the surface layer as the mixed layer is shallow in the morning and vertically well mixed during the daytime resulting in low surface NO$_2$ concentrations (Crawford et al., 2020). On the other hand, vertical column NO$_2$ concentrations show large values in the afternoon due to the continued emission of NOx.

All three emission inventories resulted in simulations that underestimated both column and surface HCHO values by up to -8.5 x 10$^{15}$ molecules·cm$^{-2}$ (-46%) at 7 LT and -0.9 ppbv (-26%) at the surface on average. The underestimations of the model HCHO relative to the Pandora and surface observations are similar to findings from comparisons of the model results with the aircraft data (Figure 9). Therefore, the model VOC performance needs to be investigated at Olympic Park.

The diurnal variations of the model O$_3$, CO, TOL, and XYL were evaluated against the surface observations at Olympic Park acquired during the KORUS-AQ campaign (Figure 11). The diurnal pattern and hourly averaged mixing ratio of O$_3$ were well

simulated with the three emission inventories with slight model negative biases. The

observed CO was 2.7 times higher than the model on average. Considering the diurnal

profile of observed TOL and XYL, KOV5 exhibited smaller negative biases than EDV2

and EDV3, but it still showed negative biases. The model TOL and XYL showed peak

concentrations at 08 LT, but the observation had a maximum value at 06 LT. The model

biases of XYL (-3.7 to -0.6 ppb, -89 to -20%) were much larger than those in TOL at

the surface. Our study demonstrates that the improvement of VOC emissions and

chemistry representations in the model is necessary for better simulations of air quality

over SMA and South Korea.

**3.4. The model performance over South Korea for the Local and Transport Cases**

Previous studies have used meteorological conditions to classify synoptic patterns that

affect air pollutant concentrations (Park et al. 2021; Peterson et al. 2019). In contrast,

we defined the Transport and Local cases by comparing model results that used the

EDV3 base emission and the EDV3 zero-out-Chinese emission (see Figure 12). The

Local case comprises May 4, May 20, June 2, and June 3 (Supporting Information,

Figure S12), while the Transport case includes May 25, May 26, and May 31

(Supporting Information, Figure S13). The Local (Transport) case in this study

generally aligns with the Stagnant and Blocking (Transport) cases in Peterson et al.

(2019); the Stagnant and Blocking is the period that a large anticyclone was located

over South Korea, and the Transport case is the period that South Korea was largely

affected by long-range transport of air pollutants by westerly wind. The Local case has

a Chinese contribution to $O_3$ of under 11%, whereas the Transport case has a Chinese

contribution to $O_3$ of over 46%. EDV3 performed better in simulating $O_3$ for the

Transport case compared to EDV2 and KOV5, with a bias of only 2.7 ppb in
comparison with the DC-8 airborne observations. In contrast, for the Local case, all
emissions had a negative bias ranging from 15.5-18.2 ppb. See the Table S11 and S12
in Supporting Information to obtain detailed information of model performances
against DC-8 measurements for different cases. Surface concentrations of $O_3$ at
Olympic Park also exhibited enhanced contributions from Chinese anthropogenic
emissions for Transport case (Figure S14). This section focuses on the model
simulations using EDV3 and its modified versions, EDV3_Ch2, EDV3_Ko2 and
EDV3_ChKo2 (doubling Chinese and South Korean CO and VOC emissions).
Figure 13 illustrates the biases in the model $O_3$, CO, and HCHO using EDV3 and
its variants relative to DC-8 observations over SMA. The plot highlights differences in
biases for the Local and Transport cases. The model $O_3$ biases were negative, and the
absolute values of biases were larger in the Local case than in the Transport case (-20%
versus -6%). The model CO biases were also negative, and the absolute values of biases
were larger in the Transport case than in the Local case. The model HCHO biases were
negative and similar for the two cases, except for a larger discrepancy between model
and observation in the Local case than in the Transport case.
Doubling Chinese CO and VOC emissions (EDV3_Ch2) only slightly reduced
biases in the Local case, whereas doubling South Korean CO and VOC emissions
(EDV_Ko2) reduced biases more compared to the EDV3_Ch2 case. Doubling South
Korean CO and VOC emissions as well as Chinese CO and VOC emissions (EDV3
ChKo2) led to the best results in $O_3$ and CO for the Local case. For the Transport case,
doubling Chinese CO and VOC emissions reduced biases to almost zero for CO and
HCHO, but the model $O_3$ was much overestimated, with 14% positive biases (from an

original bias of -6%). Doubling South Korean CO and VOC emissions reduced the biases in $O_3$ and CO a bit, but overestimated HCHO. The overestimation of $O_3$ in the Transport period in the EDV3_Ch2 and EDV3_ChKo2 cases can be explained by not only excessive ISO but also overpredicted background $O_3$ from doubled CO and VOC emissions in China (Supporting information, Table S9-S13). Doubled CO and VOC emissions overestimated $O_3$ concentrations over the Yellow Sea, which implies that the enhanced background $O_3$ increase can increase the $O_3$ level in SMA (Supporting Information, Figure S15) (Kim et al., 2023).

Further increasing South Korean CO and VOC emissions in addition to the increase of Chinese CO and VOC emissions led to overestimations of $O_3$ (20%) and HCHO (33%). These sensitivity tests modifying EDV3 indicate that increases in CO and VOC emissions over South Korea improve the model $O_3$, CO, and VOC simulations. However, increasing Chinese VOC (and CO) emissions may overestimate the model $O_3$ for the studied period.

**4. Discussion: strategy for accurate surface $O_3$ simulations over urban and regional areas in China and South Korea**

Due to unprecedentedly rich observational data sets acquired during KORUS-AQ, we investigated the status of $O_3$ simulations and outlined directions for their improvements in SMA and South Korea. In this section, strategies for the enhanced accuracy of surface $O_3$ simulations over urban and regional areas in China and South Korea are discussed. The discussion is based on the model simulations incorporating various emission scenarios derived from EDV3. In Figures 14 and 15, diverse emission cases are labeled from C1 to C7. Specifically C1, C2, and C3 correspond to EDV3_ChKo2, EDV3_Ch2,

and EDV3_Ko2, respectively. Meanwhile, C4, C5, C6, and C7 represent scenarios
involving a 50% reduction in Chinese $NO_x$ emissions, a 50% reduction in Chinese VOC
emissions, a simultaneous 50% reduction in both Chinese $NO_x$ and VOC emissions,
and a 75% reduction of Chinese $NO_x$ emissions, respectively, as discussed in Kim et al.
(2023). Examining various options involving the increase and decrease of $NO_x$ and
VOC emissions from C1 to C7 sheds light on the direction for improving $O_3$ simulations.

7        Figure 14 illustrates the model $O_3$ and $NO_2$ biases (%) in each region for all cases

based on EDV3 (Supporting Information, Table S14-S15 for details). EDV3
demonstrated good performance in simulating $O_3$ and $NO_2$ for the NCP, KOR, NEC,
and NOC region. The most substantial model $O_3$ biases were observed in SCG and SEC,
with minimal model $NO_2$ biases. Conversely, the largest model $NO_2$ biases were found
in YRD and PRD, accompanied by modest model $O_3$ biases. Improvements are needed
for model $O_3$ in SCG, SEC, YRD, and PRD with reasonable $NO_2$ simulations. For SCG
and SEC, the C5 case (50% VOC emission reduction only) exhibited the lowest $O_3$
biases. Doubled Chinese VOC emission case (C1 and C2) in SCG and SEC resulted in
increased $O_3$ biases to ~100%, compared to 68% in the EDV3 case. In this study, the
anthropogenic VOC emissions were reduced. Further reductions of biogenic VOC
emissions as well as anthropogenic emissions need to be explored in the future. For
SCG and SEC, a reduction in $NO_x$ emissions also led to a slight decrease in $O_3$ biases.
FNR values for the two regions are about 1.3, which turned out to be still VOC-limited
or in a transitional state. For the YRD and PRD regions, first, $NO_x$ emissions need to
be reduced to improve $NO_2$ biases in the model. The case C6 (50% reductions in both
NOx and VOC emissions) yielded the most favorable $O_3$ and $NO_2$ simulations. Solely
reducing $NO_x$ emissions (as in case C4) increase $O_3$ biases by 26-37% relative to EDV3.
The FNR values for YRD and PRD are 0.32 and 0.52, respectively, placing them in the
VOC-limited regime (FNR < 1). In general, an increase in Chinese VOC emissions (as
observed in cases C1 and C2) resulted in elevated surface ozone levels for all regions,
including KOR. For NCP, KOR, NEC, and NOC where the model $O_3$ and $NO_2$ agree
with the observations, reducing VOC proves to be an effective strategy for decreasing
surface $O_3$. The biases of $O_X$ typically follow $O_3$ biases across cases in all regions except
NCP, YRD, PRD, and NOC, which experience high $NO_X$ conditions. Refer to
Supporting Information Figure S16 for analysis of $O_X$ along with $O_3$ across various
regions. In these specific regions, a substantial reduction in $NO_X$ levels (as in C4 and
C7) resulted in an increase in $O_3$ bias, while there was a decrease in $O_X$.

11       In Figure 15, the model $O_3$ and $NO_2$ biases (%) in the 12 megacities in China and

South Korea are illustrated for all cases. Refer to Supporting Information Table S16 and
S17 for specific values. EDV3 showed effective performance in simulating $O_3$ and $NO_2$
for cities such as Beijing, Tianjin, Hangzhou, SMA, and Xian. The most substantial
model $O_3$ biases were observed in Chengdu and Chongqing, with minimal model $NO_2$
biases. In contrast, the notable model $NO_2$ biases were identified in Shanghai, Nanjing,
Guangzhou, Shenzhen, and Wuhan, accompanied by modest model $O_3$ biases. For
Chengdu and Chongqing, situated roughly in SCG, the C5 case (50% VOC emission
reduction only) results in the lowest $O_3$ biases with decent $NO_2$ simulations. For
Shanghai, Nanjing, Guangzhou, Shenzhen, and Wuhan, case C6 (50% reductions in
both $NO_X$ and VOC emissions) produced the most favorable $O_3$ and $NO_2$ simulations.
Simply reducing $NO_X$ emissions (as in case C4) increase $O_3$ biases in these cities.
Overall, the increase in Chinese VOC emissions (cases C1 and C2) resulted in elevated
surface ozone levels for all cities, including SMA with an increase in biases, except for

Shanghai. Reduction of only VOC emissions (C5) led to the lowest surface $O_3$ levels for all cities. The biases of $O_X$ generally follow $O_3$ biases in Chengdu and Chongqing, where the simulated $O_3$ initially exhibits a notably high positive bias (50-60%), attributable to high VOC. Refer to Supporting Information Figure S17 for an analysis of $O_x$ and $O_3$ across cases and cities. In contrast, for other cities experiencing high $NO_X$ conditions with positive $NO_2$ biases, a reduction in $NO_X$ levels (as in C4 and C7) led to a decrease in $O_X$ (and its bias for most cities). However, there was a simultaneous increase in $O_3$ and its bias, attributed to the $NO_x$-saturated regime (Figure S17).

## 5. Summary and conclusions

We conducted sensitivity tests using WRF-Chem with three different bottom-up emission inventories (EDGAR-HTAP v2, v3, and KORUS v5) to investigate the impacts of different emissions on the simulation of $O_3$ and precursors in East Asia. This study is the first to use EDGAR-HTAP v3 with WRF-Chem v4.4 and extends the validation domain to the whole of China during the KORUS-AQ campaign period. We extensively evaluated these emission inventories using both ground and aircraft observations in East Asia.

The three emission inventories accurately reproduced the diurnal profiles and absolute values of surface $O_3$ for most subregions in China, except for the SCG and SEC areas. However, discrepancies were observed in the model performance for the MDA8 $O_3$ concentrations, with poor correlations observed over regions with high HCHO concentrations (south of 30°N) and relatively low ratios of FNR (north of 30°N). The emission inventories reasonably reproduced the spatial distribution of daily surface

NO$_2$ concentrations. However, we found that CO was considerably underestimated by
the emission inventories over both China and South Korea.

3       We evaluated the model simulations against vertical profile measurements of O$_3$,

NO$_2$, CO, HCHO, TOL, XYL, ETE, and ISO from the DC-8 aircraft, as well as surface
observations over South Korea. The simulated vertical shapes of O$_3$, NO$_2$, CO, HCHO,
TOL, XYL, ETE, and ISO agreed well with the DC-8 measurements in the SMA,
although negative biases were observed for O$_3$, CO, TOL, XYL, and HCHO, with the
largest discrepancy between the model results and observations in the Chungnam area.
When we compared the simulations with the surface in-situ measurements and
PANDORA observations at the Olympic Park in Seoul, the model accurately
reproduced the diurnal patterns of surface and vertical columns of NO$_2$ and HCHO.
However, we found that the model underestimated TOL and XYL. This underestimation
of TOL and XYL is one of the reasons why the model underestimates O$_3$ concentrations,
as VOCs contribute to NO to NO$_2$ conversions resulting in O$_3$ production via
photochemistry.

16       We also classified the flight tracks into two categories: Local and Transport cases.

We found that the negative bias of O$_3$ was much larger under the Local case than the
Transport case. When the increment of CO and VOC emissions in South Korea is taken
into account, the biases of O$_3$ are significantly reduced, indicating the need for local
emission adjustments to decrease O$_3$ bias in South Korea.

21       To improve surface O$_3$ simulations over China and South Korea using EDV3,

lowering VOC emissions are advantageous for SCG and SEC including urban areas
like Chengdu and Chongqing. Meanwhile, for YRD and PRD regions, as well as cities
such as Shanghai, Nanjing, Guangzhou, Shenzhen, and Wuhan, both NOx and VOC

emissions should be reduced to enhance model performances. Increasing VOC

emissions adversely affected the model's accuracy in simulating $O_3$ in China, leading

to increased biases.

Our study revealed a consistent overestimation of isoprene over SMA. The

uncertainty of biogenic VOC emissions from MEGAN can affect the model $O_3$

performance. Therefore, to achieve more accurate simulations of $O_3$ in East Asia, it is

essential to explore precise representations of both anthropogenic and biogenic VOC

emissions.

**Code and data availability**

WRF-Chem source codes are distributed by NCAR

(https://doi:10.5065/D6MK6B4K). WRF-Chem v4.4 is available in the GitHub (wrf-

model, 2022). The exact version of WRF-Chem codes and configuration files are

archived at https://doi.org/10.5281/zenodo.8260026 (Kim et al., 2023). National

Centers for Environmental Prediction (NCEP) FNL data can be accessed from Research

Data Archive (RDA) (NCEP, 2019). The CAM-chem data for boundary conditions is

also obtained from RDA (ACOM, 2019; doi.org/10.5065/CKR4-GP38). The EDGAR-

HTAP v2 data can be downloaded in the website

(https://edgar.jrc.ec.europa.eu/dataset_htap_v2). The EDGAR-HTAP v3 is archived on

Zenodo (Crippa, 2023). The KORUS-AQ data are available from the website

(https://www-air.larc.nasa.gov/cgi-bin/ArcView/korusaq)

(doi:10.5067/Suborbital/KORUSAQ/DATA01). The EDGAR-HTAP v2, v3, and

KORUS v5 data including emission processing programs are available at

https://doi.org/10.5281/zenodo.8260026 (Kim et al., 2023).

**Author contribution**

KMK conducted simulations, analysis and wrote the paper. SWK designed this study, secured funding, performed analysis and wrote the paper. SS supported model set-up and contributed to refining the paper. DRB measured VOC data from DC-8. SC acquired ground-based in-situ VOC data at Olympic Park. JHC performed analysis and wrote the paper. LKE and GGP assisted in setting up the model emissions and discussed about the model performance. AF measured HCHO data from DC-8. JRH measured Pandora data ($NO_2$, HCHO). JH retrieved PBL height and discussed about the results. JJ acquired $NO_2$ data at Olympic Park with different methods. AJW acquired $NO_2$ and $O_3$ data from DC-8. JHW and QZ provided emissions inventories and related information. All authors reviewed the manuscript.

**Competing interests**

At least one of the (co-)authors is a member of the editorial board of Geoscientific Model Development.

**Acknowledgements**

This work was supported by the National Research Foundation of Korea (NRF) grant funded by the Korea government (MSIT) (No. 2020R1A2C2014131). S.-W. Kim also acknowledges support from NRF-2018R1A5A1024958. All the computing resources are provided by National Center for Meteorological Supercomputer. The National Center for Atmospheric Research (NCAR) is sponsored by the National Science Foundation (NSF) (NNX16AD96G). We would like to express our gratitude to Glen

Diskin for generously providing the CO data from the DC-8 aircraft. We also thanks to Andrew Whitehill and Russell Long for providing the HCHO data from Olympic Park. We would also like to thank Meng Li and Brian McDonald for their valuable discussions, which greatly enhanced our understandings.

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

**Table List**

Table 1. The model experiments with different emissions.

Table 2. Comparison of the ground-based hourly $O_3$, $NO_2$, and CO observations with the simulations utilizing EDGAR-HTAP v2 (EDV2) and v3 (EDV3) and KORUS v5 (KOV5) in each regional box (unit = ppb). N is the number of samples. R is correlation coefficient.

Table 3. Comparison of total NOx, TOL, XYL, biogenic isoprene emissions, and formaldehyde to $NO_2$ ratio (FNR) for different emission data sets in each regional box. The MEGAN biogenic isoprene emissions are equally applied to all simulations using different emission data. (unit = mol/s for emissions)

Table 4. Comparison of aircraft-based 1-minuite-interval $O_3$, $NO_2$, CO, HCHO, TOL, XYL, ETE, and ISO observations with EDV2, EDV3, and KOV5 for all flight cases under 2 km height (unit = ppb). N is the number of samples. R is correlation coefficient.

**Figure List**

Figure 1. The averaged spatial distribution map of the NO, CO, and TOL (toluene + less reactive aromatics) emissions from EDGAR-HTAP v2, v3, and KORUS v5 in May. The boxes represent Northern China (NOC, 38-42˚N/106-110˚E), Sichuan-Chongqing-Guizhou (SCG, 27-33˚N/103-109˚E), Pearl River Delta (PRD, 21.5-24˚N/112-115.5˚E), Southeastern China (SEC, 24-28˚N/116-120˚E), Yangtze River Delta (YRD, 30-33˚N/119-122˚E), South Korea (KOR, 34.5-38˚N/126-130˚E), North China Plain (NCP, 34-41˚N/113-119˚E), and Northeastern China (NEC, 43-47˚N/124-130˚E). NOC, NEC, and SEC are denoted by blue boxes (non-urban). NCP, SCG, PRD, YRD, and KOR are denoted by red boxes (urban).

Figure 2. The DC-8 flight paths during the KORUS-AQ campaign period (black) and 6 regional boxes (1: Seoul Metropolitan Area (SMA); 2: Yellow Sea; 3: Chungnam; 4: Kyungbuk; 5: Gwangju; 6: Busan) (red).

Figure 3. Averaged $O_3$ concentrations from ground-based observations and model simulations over the areas that distinguish urban (red box) and non-urban (green box) region (central plot). Box-averaged diurnal cycle (solid lines) of $O_3$ and 1/4 of standard deviations (filled area) from observations (black), EDV2 (sky blue), EDV3 (blue), and KOV5 (red) by local time are shown. The results are shown for NOC, SCG, PRD, SEC, YRD, KOR, NCP, and NEC.

Figure 4. Comparison of (a) the campaign averaged ground-based maximum daily average of 8-hour $O_3$ (MDA8 $O_3$) (unit: ppb) observations and WRF-Chem simulations with (d) EDGAR-HTAP v2 (EDV2), (e) v3 (EDV3), (f) KORUS v5 (KOV5) and (g, h, i) the differences between the observations and model results. The sub-regions are presented with red (urban) and green (non-urban) boxes. The scatter plots comparing averaged observations and the three-emission-based WRF-Chem simulations (sky blue; EDV2, blue; EDV3, red; KOV5) are shown in (b) and (c) for Eastern China and South Korea, respectively. (a, d-e) Color-filled circles in (a), (d), (e), and (f) represent the averaged MDA8 $O_3$ for the whole campaign period (1st May to 10th June).

Figure 5. The same as Figure 3 except $NO_2$.

Figure 6. The same as Figure 4 except daily $NO_2$ (unit: ppb).

Figure 7. The same as Figure 4 except daily CO (unit: ppm).

Figure 8. The mean (bars) and 1/4 of standard deviations (whiskers) of (a) $O_3$, (b) $NO_2$, (c) CO, (d) HCHO, (e) TOL, (f) XYL, (g) ethene (ETE), and (h) isoprene (ISO) (unit = ppb) from DC-8 (dark grey), EDV2 (sky blue), EDV3 (blue), and KOV5 (red) for each box are shown, respectively. TOL and XYL are calculated based on Table S8 (Supporting Information). The contribution of toluene to TOL and m/p-Xylene + o-Xylene to XYL is represented with light grey bars (e, f). The sampling numbers are represented with magenta color above the plots.

Figure 9. Vertically averaged (a) $O_3$, (b) $NO_2$, (c) CO, (d) HCHO, (e) TOL, (f) XYL, (g) ETE, and (h) ISO from DC-8 (black), EDV2 (sky blue), EDV3 (blue), and KOV5 (red) in SMA under 2 km height above ground level. The 1/2 of standard deviations are represented with black whiskers in each 200m layer. The sample number is presented with magenta color on the right side of the plots.

Figure 10. The diurnal cycles of vertical columns and surface concentrations of (a) $NO_2$ and (b) HCHO from Pandora spectrometer (column), and ground-based instruments (TEI 42i NOx analyzer and Aerodyne QCL) at the Olympic Park site (37.5232˚N, 127.126˚E). EDV2 (sky blue), EDV3 (blue), and KOV5 (red) are compared with observations. The WRF-Chem vertical column concentrations are produced by summing all vertical layers.

Figure 11. Diurnal cycles of surface (a) $O_3$, (b) CO, (c) TOL, and (d) XYL at the Olympic Park site. EDV2 (sky blue), EDV3 (blue), and KOV5 (red) are compared with the observations. 1/4 of standard deviations are represented with grey shades. The average period is from the 11th May to the 10th June.

Figure 12. Averaged $O_3$ (bars) and 1/4 of standard deviations (whiskers) (unit: ppbv) for the 20 DC8 flights (under 2 km height). The observations (grey) are compared with the model results utilizing EDV2 (sky blue), EDV3 (blue), and KOV5 (red). White hatch-filled bars over blue bars are the contribution of Chinese emissions to $O_3$ concentrations obtained from the default and sensitivity model runs with/without Chinese anthropogenic emissions. The Local (5/4,20 and 6/2,3) and Transport (5/25,26,31) cases are shaded with light blue and orange, respectively.

Figure 13. The biases in (a) the model $O_3$, (b) CO, and (c) HCHO concentrations (bars) relative to the DC-8 observations under 2 km height over SMA (dark gray: EDV3, red:

EDV3 Ch2, blue: EDV3 ChKo2): (left panel) Local and (right panel) Transport case. Fractional differences (%) are shown in the white boxes.

Figure 14. Comparison of relative biases ((Model-Observation)/Observation, unit=%) of daily $O_3$ and $NO_2$ at surface observation sites during the KORUS-AQ campaign period from sensitivity simulation (C1-7) with EDV3 in each region (NCP, SCG, YRD, PRD, KOR, NEC, NOC, and SEC). C1; EDGAR-HTAP v3 with double CO and VOC emission in China and South Korea, C2; EDGAR-HTAP v3 with double CO and VOC emission in China, C3; EDGAR-HTAP v3 with double CO and VOC emission in South Korea, C4; EDGAR-HTAP v3 with 50% NOx reduction in China, C5; EDGAR-HTAP v3 with 50% VOC reduction in China, C6; EDGAR-HTAP v3 with 50% NOx and VOC reduction in China, C7; EDGAR-HTAP v3 with 75% NOx reduction in China.

Figure 15. Same as Figure 14 except that the region is changed to cities; Beijing (39.4-41.1N, 115.4-117.5E), Tianjin (38.55-40.25N, 116.7-118.1E), Chengdu (30.05-31.5N, 103-105E), Chongqing (28.15-32.25N, 105.3-110.2E), Shanghai (30.7-31.5N, 120.85-122E), Hangzhou (29.2-30.6N, 118.3-120.9E), Nanjing (31.2-32.65N, 118.35-119.25E), Guangzhou (22.55-24N, 112.9-114.05E), Shenzhen (22.4-22.9N, 113.7-114.65E), SMA (37.2-37.8N, 126.5-127.3E), Wuhan (29.95-31.4N, 113.65-115.1E), and Xian (33.65-34.75N, 107.65-109.9E).

1    **Table 1**. The model experiments with different emissions.

| Experiments | Emissions |
| --- | --- |
| EDV2 | EDGAR-HTAP v2 |
| EDV3 | EDGAR-HTAP v3 |
| KOV5 | KORUS v5 |
| EDV3_Ch2 | EDGAR-HTAP v3 with double CO, VOC emission in China |
| EDV3_Ko2 | EDGAR-HTAP v3 with double CO, VOC emission in South Korea |
| EDV3_ChKo2 | EDGAR-HTAP v3 with double CO, VOC emission in China & South Korea |

**Table 2.** Comparison of the ground-based hourly O₃, NO₂, and CO observations with
the simulations utilizing EDGAR-HTAP v2 (EDV2) and v3 (EDV3) and KORUS v5
(KOV5) in each regional box (unit = ppb). N is the number of samples. R is correlation
coefficient.

| Region | | [1]NCP | [1,a]SCG | [1]YRD | [1]PRD | [1,b]KOR (SMA) | [2,c]NEC | [2,d]NOC | [2,e]SEC |
|---|---|---|---|---|---|---|---|---|---|
| **N** | | 190 | 104 | 93 | 68 | 358 (125) | 45 | 28 | 43 |
| **O₃** OBS | Mean | 44.5 | 34.6 | 38.2 | 27.9 | 41.5 (36.6) | 40.9 | 44.3 | 26.1 |
| | Mean | 32.2 | 53.5 | 21.6 | 27.6 | 40.5 (31.1) | 28.6 | 39.4 | 40.8 |
| EDV2 | Bias | -12.3 | 18.9 | -16.6 | -0.3 | -1.0 (-5.5) | -12.3 | -4.9 | 14.7 |
| | R | 0.65 | 0.53 | 0.62 | 0.61 | 0.59 (0.60) | 0.48 | 0.63 | 0.52 |
| | Mean | 43.4 | 57.5 | 35.7 | 34.7 | 41.0 (32.6) | 35.2 | 43.7 | 45.5 |
| EDV3 | Bias | -1.1 | 23.0 | -2.5 | 6.8 | -0.5 (-4.0) | -5.7 | -0.6 | 19.4 |
| | R | 0.68 | 0.55 | 0.66 | 0.65 | 0.56 (0.57) | 0.63 | 0.67 | 0.55 |
| | Mean | 49.0 | 55.3 | 41.1 | 35.7 | 42.2 (33.1) | 37.1 | 43.8 | 42.4 |
| KOV5 | Bias | 4.5 | 20.7 | 2.8 | 7.8 | 0.7 (-3.5) | -3.8 | -0.5 | 16.3 |
| | R | 0.71 | 0.53 | 0.65 | 0.70 | 0.62 (0.64) | 0.62 | 0.67 | 0.54 |
| **NO₂** OBS | Mean | 17.5 | 13.8 | 17.1 | 12.9 | 23.2 (32.5) | 13.5 | 11.9 | 9.6 |
| | Mean | 25.8 | 12.7 | 39.8 | 22.0 | 18.8 (29.6) | 13.7 | 12.9 | 11.0 |
| EDV2 | Bias | 8.3 | -1.0 | 22.7 | 9.1 | -4.3 (-3.0) | 0.2 | 1.0 | 1.5 |
| | R | 0.45 | 0.37 | 0.38 | 0.54 | 0.51 (0.34) | 0.49 | 0.47 | 0.19 |
| | Mean | 21.8 | 12.2 | 30.4 | 21.0 | 21.3 (31.8) | 11.2 | 10.3 | 11.3 |
| EDV3 | Bias | 4.3 | -1.6 | 13.3 | 8.1 | -1.9 (-0.8) | -2.3 | -1.6 | 1.7 |
| | R | 0.44 | 0.34 | 0.36 | 0.52 | 0.49 (0.31) | 0.49 | 0.52 | 0.22 |
| | Mean | 13.9 | 7.5 | 23.5 | 13.3 | 17.7 (28.3) | 7.0 | 7.7 | 7.7 |
| KOV5 | Bias | -3.6 | -6.3 | 6.4 | 0.3 | -5.5 (-4.2) | -6.5 | -4.2 | -1.9 |
| | R | 0.44 | 0.37 | 0.41 | 0.52 | 0.51 (0.39) | 0.49 | 0.51 | 0.26 |
| **CO** OBS | Mean | 835 | 597 | 694 | 636 | 443 (493) | 527 | 579 | 655 |
| | Mean | 373 | 389 | 455 | 282 | 175 (210) | 206 | 162 | 258 |
| EDV2 | Bias | -462 | -208 | -239 | -354 | -267 (-283) | -321 | -417 | -397 |
| | R | 0.24 | 0.20 | 0.42 | 0.30 | 0.31 (0.30) | 0.21 | 0.09 | 0.18 |
| | Mean | 374 | 359 | 535 | 282 | 196 (208) | 221 | 162 | 256 |
| EDV3 | Bias | -461 | -238 | -159 | -354 | -247 (-285) | -306 | -417 | -398 |
| | R | 0.22 | 0.19 | 0.35 | 0.31 | 0.26 (0.33) | 0.24 | 0.10 | 0.25 |
| | Mean | 355 | 358 | 475 | 305 | 190 (217) | 231 | 176 | 266 |
| KOV5 | Bias | -480 | -239 | -219 | -331 | -253 (-276) | -296 | -404 | -388 |
| | R | 0.27 | 0.21 | 0.48 | 0.29 | 0.35 (0.36) | 0.15 | 0.10 | 0.13 |

1) Urban area, 2) Non-urban area
a) Sichuan-Chongqing-Guizhou, b) South Korea, c) Northeastern China, d) Northern China, e) Southeastern China
**Table 3.** Comparison of total NOx, TOL, XYL, biogenic isoprene emissions, and
formaldehyde to NO$_2$ ratio (FNR) for different emission data sets in each regional box.
The MEGAN biogenic isoprene emissions are equally applied to all simulations using
different emission data. (unit = mol/s for emissions)

| Type | emissions | NCP | SCG | YRD | PRD | KOR(SMA) | NEC | NOC | SEC |
|---|---|---|---|---|---|---|---|---|---|
| **NOx emission** | **EDV2** | 5967 | 1500 | 2366 | 1178 | 990(196) | 987 | 688 | 590 |
| | **EDV3** | 5202 | 1654 | 1642 | 1091 | 1191(214) | 876 | 597 | 662 |
| | **KOV5** | 3237 | 902 | 1166 | 607 | 886(191) | 513 | 373 | 410 |
| **TOL emission** | **EDV2** | 140 | 56 | 84 | 47 | 27(6) | 26 | 8 | 20 |
| | **EDV3** | 220 | 77 | 99 | 68 | 27(8) | 40 | 9 | 36 |
| | **KOV5** | 403 | 106 | 234 | 155 | 98(26) | 68 | 21 | 79 |
| **XYL emission** | **EDV2** | 84 | 34 | 51 | 28 | 15(4) | 15 | 4 | 12 |
| | **EDV3** | 132 | 46 | 60 | 41 | 16(4) | 24 | 6 | 22 |
| | **KOV5** | 133 | 35 | 79 | 52 | 41(9) | 21 | 7 | 26 |
| **Biogenic isoprene emission** | | 132 | 364 | 43 | 127 | 135(6) | 106 | 23 | 310 |
| **FNR (14-16LT)** | **EDV2** | 0.25 | 1.31 | 0.19 | 0.52 | 0.53(0.19) | 0.68 | 0.76 | 1.18 |
| | **EDV3** | 0.44 | 1.30 | 0.32 | 0.52 | 0.43(0.18) | 0.93 | 0.94 | 1.33 |
| | **KOV5** | 0.72 | 2.33 | 0.48 | 1.00 | 0.71(0.22) | 1.44 | 1.49 | 1.91 |

1   **Table 4.** Comparison of aircraft-based 1-minuite-interval $O_3$, $NO_2$, CO, HCHO, TOL,

2   XYL, ETE, and ISO observations with EDV2, EDV3, and KOV5 for all flight cases

3   under 2 km height (unit = ppb). N is the number of samples. R is correlation coefficient.

| Species | Type | N | Mean | Bias | σ | R |
|---|---|---|---|---|---|---|
| $O_3$ | **OBS** | 5191 | 84.4 | | 19.9 | |
| | **EDV2** | | 67.5 | -16.8 | 16.7 | 0.44 |
| | **EDV3** | | 69.3 | -15.1 | 17.8 | 0.43 |
| | **KOV5** | | 66.9 | -17.5 | 15.8 | 0.50 |
| $NO_2$ | **OBS** | 5047 | 2.19 | | 4.49 | |
| | **EDV2** | | 3.06 | 0.87 | 4.60 | 0.71 |
| | **EDV3** | | 3.91 | 1.72 | 5.34 | 0.67 |
| | **KOV5** | | 2.83 | 0.64 | 4.73 | 0.73 |
| CO | **OBS** | 5575 | 253 | | 100 | |
| | **EDV2** | | 148 | -105 | 48 | 0.60 |
| | **EDV3** | | 156 | -97 | 47 | 0.59 |
| | **KOV5** | | 146 | -107 | 43 | 0.62 |
| HCHO | **OBS** | 5365 | 2.37 | | 1.64 | |
| | **EDV2** | | 1.75 | -0.62 | 1.01 | 0.69 |
| | **EDV3** | | 1.78 | -0.59 | 1.02 | 0.67 |
| | **KOV5** | | 1.80 | -0.57 | 1.10 | 0.71 |
| TOL | **OBS** | 730 | 2.60 | | 2.02 | |
| | **EDV2** | | 0.47 | -2.13 | 0.38 | 0.39 |
| | **EDV3** | | 0.55 | -2.05 | 0.48 | 0.38 |
| | **KOV5** | | 1.58 | -1.01 | 1.30 | 0.37 |
| XYL | **OBS** | 289 | 0.73 | | 0.65 | |
| | **EDV2** | | 0.23 | -0.50 | 0.23 | 0.30 |
| | **EDV3** | | 0.30 | -0.43 | 0.31 | 0.30 |
| | **KOV5** | | 0.49 | -0.24 | 0.47 | 0.27 |
| ETE | **OBS** | 2573 | 0.42 | | 1.59 | |
| | **EDV2** | | 0.51 | 0.09 | 0.65 | 0.14 |
| | **EDV3** | | 0.56 | 0.14 | 0.76 | 0.15 |
| | **KOV5** | | 0.51 | 0.08 | 0.58 | 0.20 |
| ISO | **OBS** | 1294 | 0.08 | | 0.09 | |
| | **EDV2** | | 0.18 | 0.10 | 0.21 | 0.41 |
| | **EDV3** | | 0.19 | 0.11 | 0.20 | 0.41 |
| | **KOV5** | | 0.17 | 0.10 | 0.20 | 0.42 |

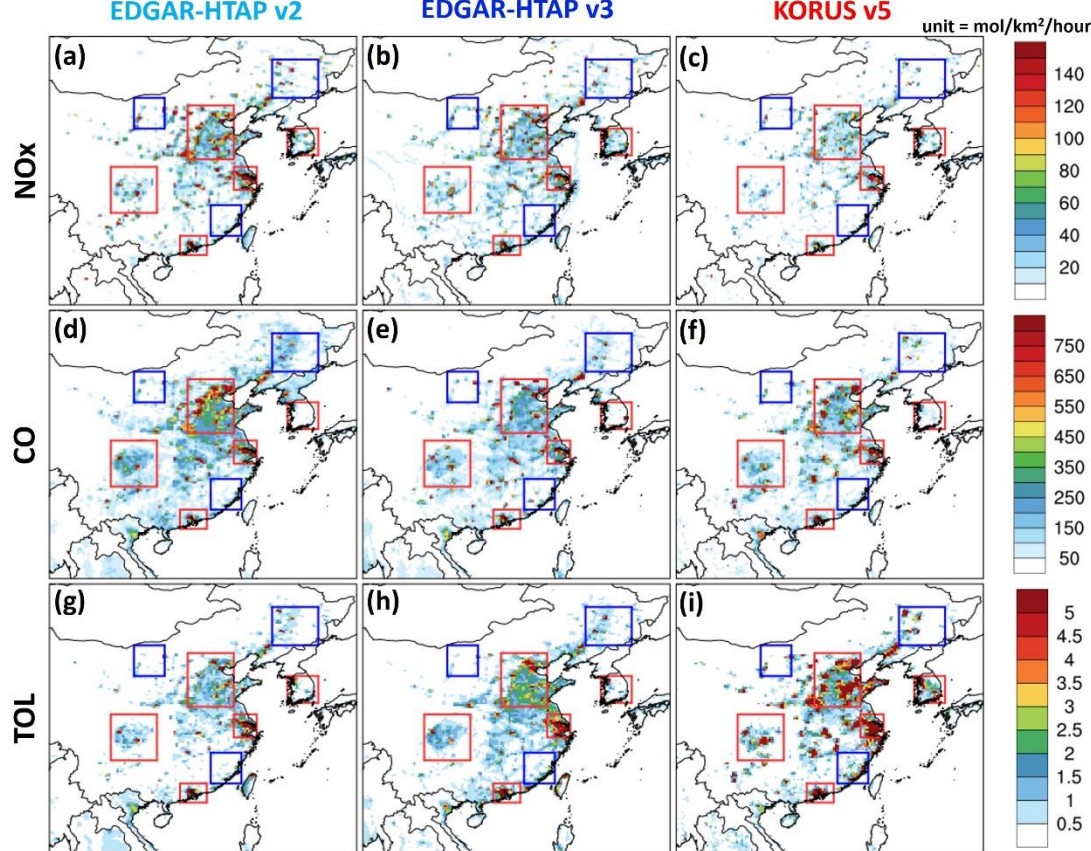

**Figure 1.** The averaged spatial distribution map of the NO, CO, and TOL (toluene +
less reactive aromatics) emissions from EDGAR-HTAP v2, v3, and KORUS v5 in May.
The boxes represent Northern China (NOC, 38-42˚N/106-110˚E), Sichuan-Chongqing-
Guizhou (SCG, 27-33˚N/103-109˚E), Pearl River Delta (PRD, 21.5-24˚N/112-115.5˚E),
Southeastern China (SEC, 24-28˚N/116-120˚E), Yangtze River Delta (YRD, 30-
33˚N/119-122˚E), South Korea (KOR, 34.5-38˚N/126-130˚E), North China Plain (NCP,
34-41˚N/113-119˚E), and Northeastern China (NEC, 43-47˚N/124-130˚E). NOC, NEC,
and SEC are denoted by blue boxes (non-urban). NCP, SCG, PRD, YRD, and KOR are
denoted by red boxes (urban).

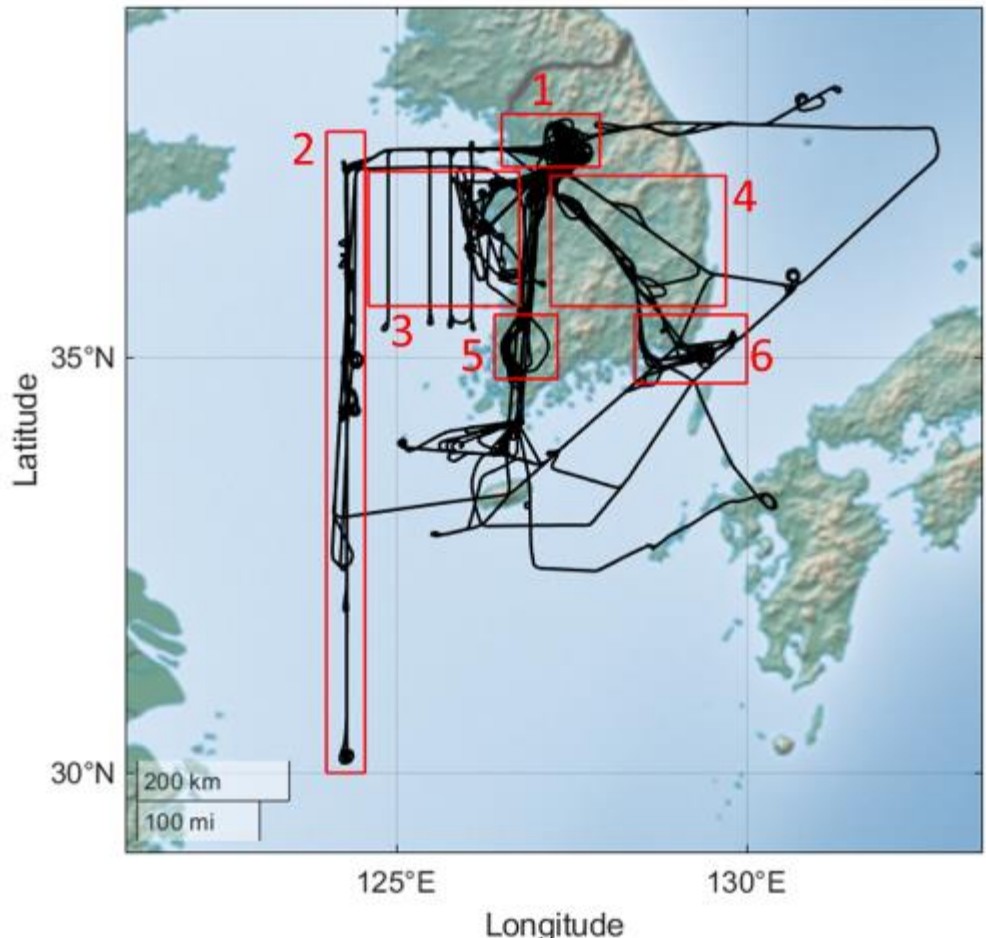

2  **Figure 2**. The DC-8 flight paths during the KORUS-AQ campaign period (black) and
3  6 regional boxes (1: Seoul Metropolitan Area (SMA); 2: Yellow Sea; 3: Chungnam; 4:
4  Kyungbuk; 5: Gwangju; 6: Busan) (red).

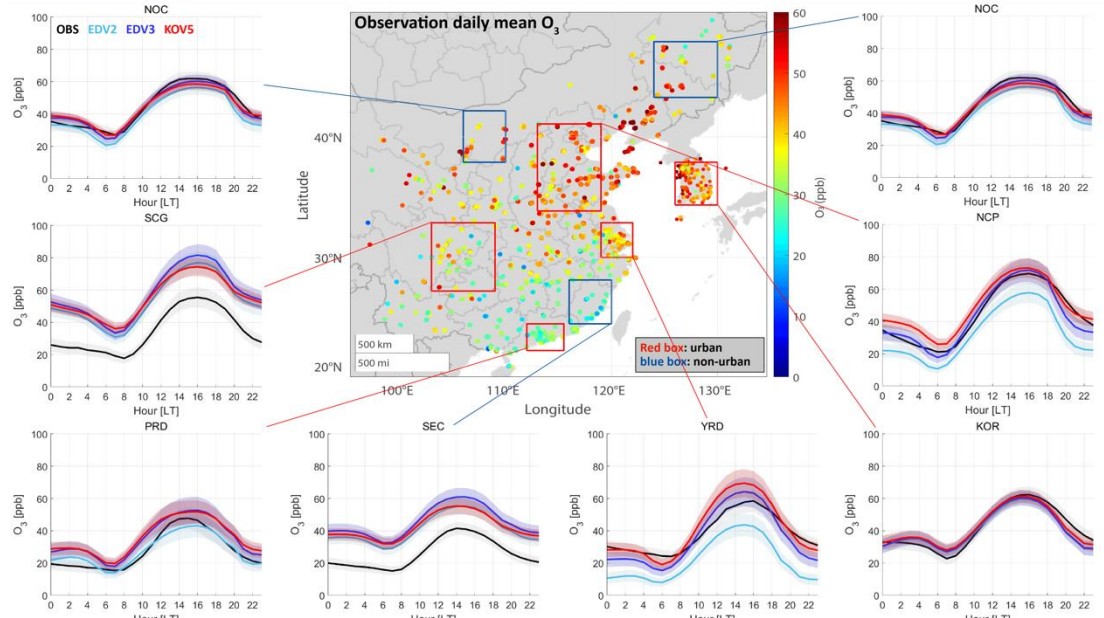

**Figure 3.** Averaged O₃ concentrations from ground-based observations and model
simulations over the areas that distinguish urban (red box) and non-urban (green box)
region (central plot). Box-averaged diurnal cycle (solid lines) of O₃ and 1/4 of standard
deviations (filled area) from observations (black), EDV2 (sky blue), EDV3 (blue), and
KOV5 (red) by local time are shown. The results are shown for NOC, SCG, PRD, SEC,
YRD, KOR, NCP, and NEC.

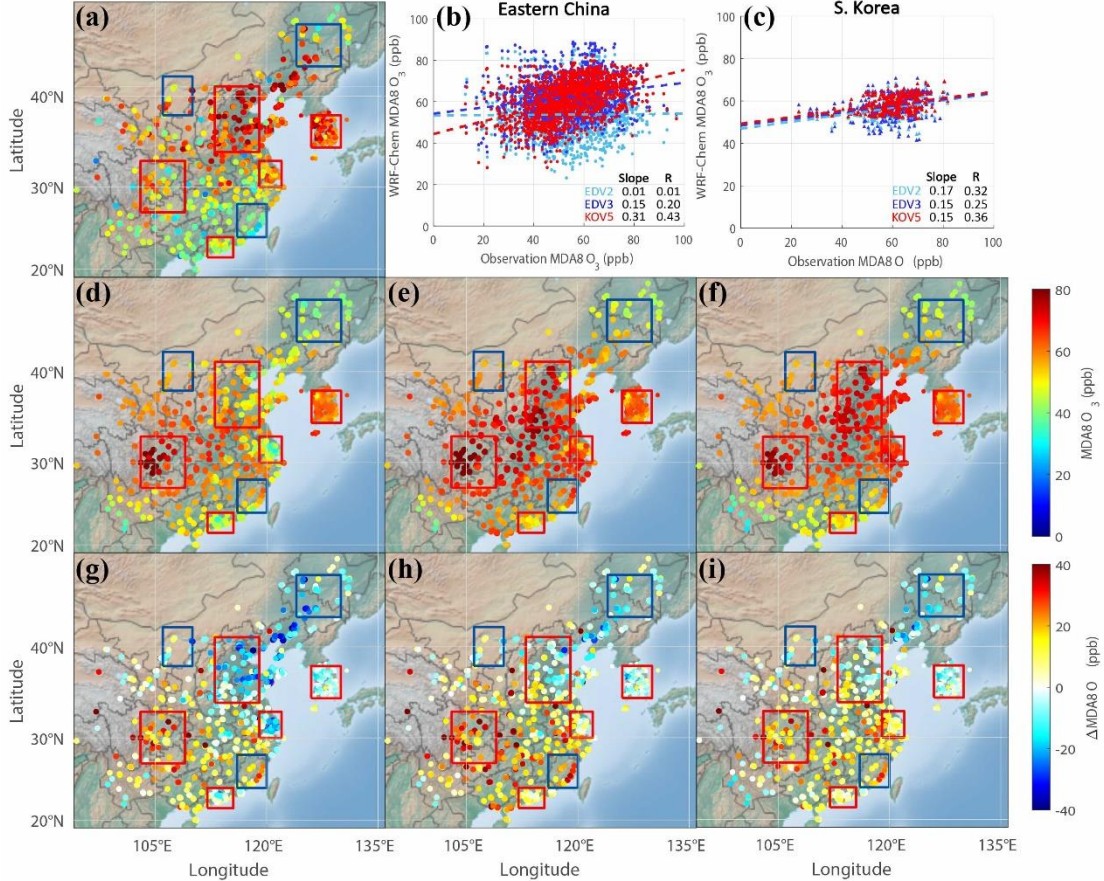

Figure 4. Comparison of (a) the campaign averaged ground-based maximum daily average of 8-hour $O_3$ (MDA8 $O_3$) (unit: ppb) observations and WRF-Chem simulations with (d) EDGAR-HTAP v2 (EDV2), (e) v3 (EDV3), (f) KORUS v5 (KOV5) and (g, h, i) the differences between the observations and model results. The sub-regions are presented with red (urban) and green (non-urban) boxes. The scatter plots comparing averaged observations and the three-emission-based WRF-Chem simulations (sky blue; EDV2, blue; EDV3, red; KOV5) are shown in (b) and (c) for Eastern China and South Korea, respectively. (a, d-e) Color-filled circles in (a), (d), (e), and (f) represent the averaged MDA8 $O_3$ for the whole campaign period (1st May to 10th June).

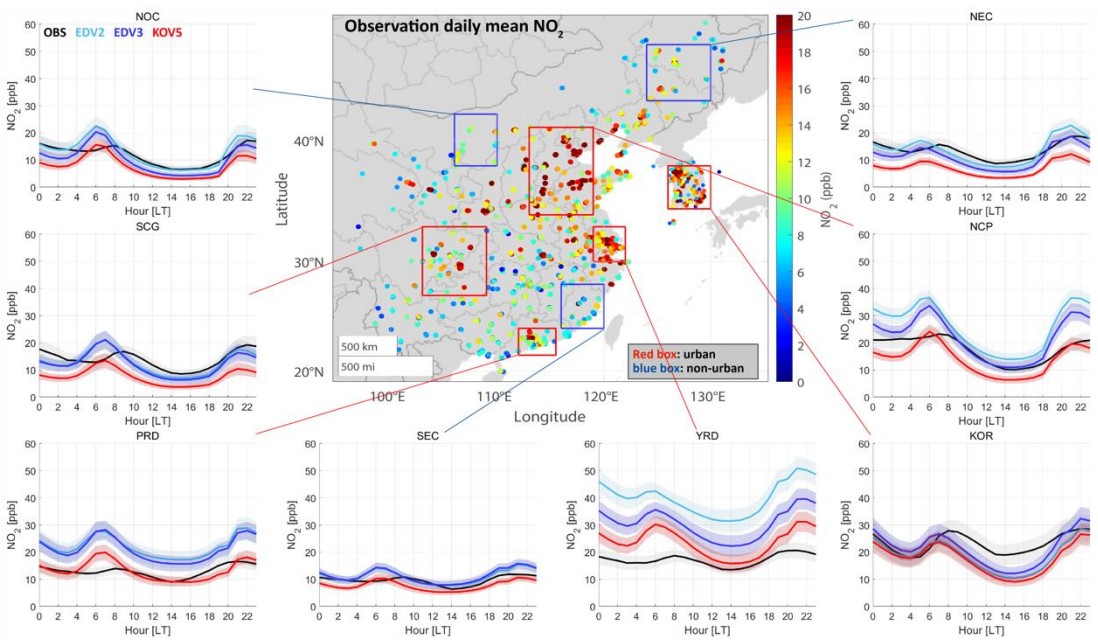

2      **Figure 5.** The same as **Figure 3** except NO₂.

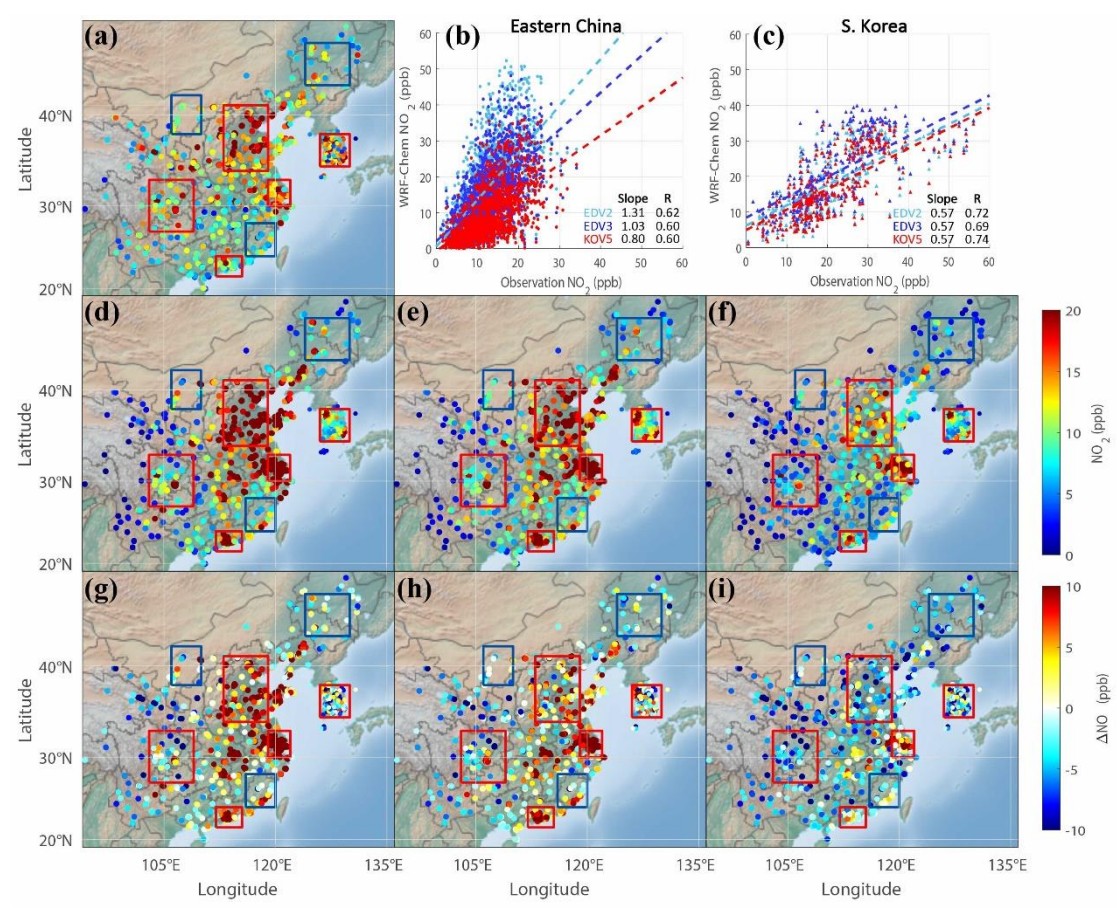

Figure 6. The same as **Figure 4** except daily NO₂ (unit: ppb).

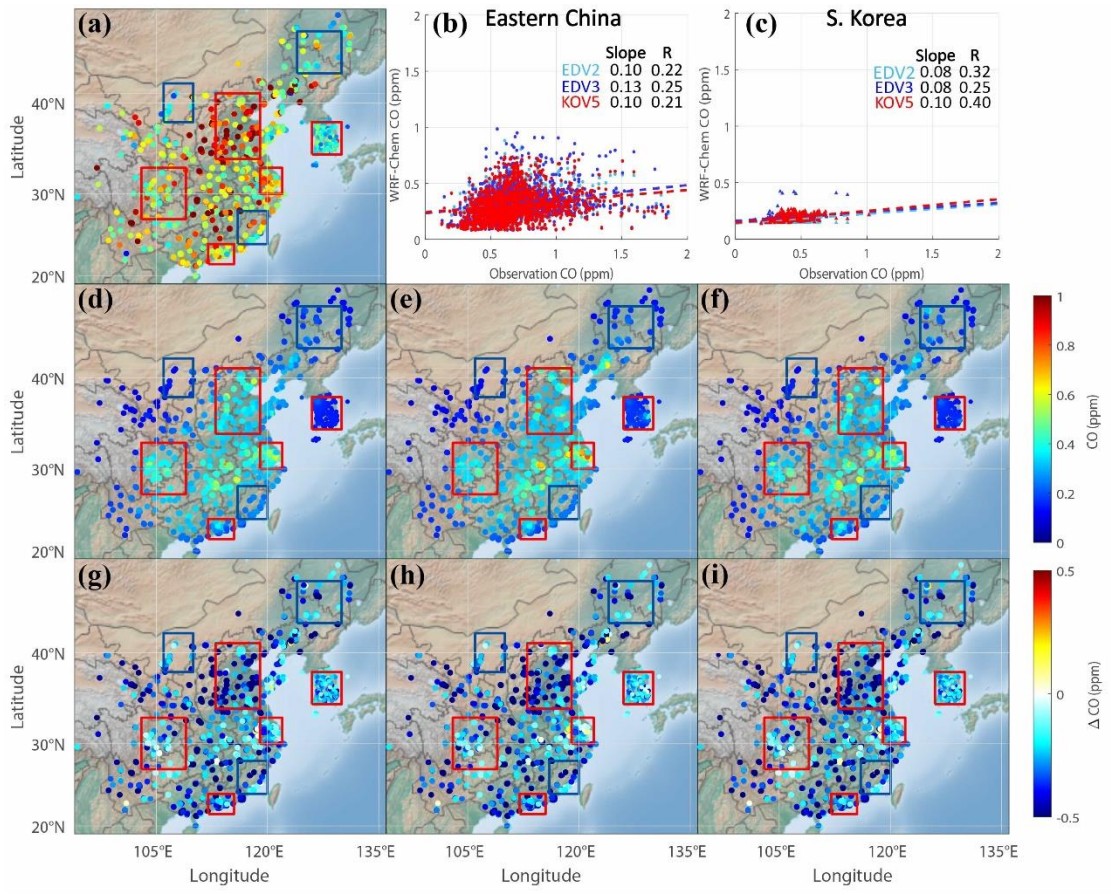

2 **Figure 7.** The same as **Figure 4** except daily CO (unit: ppm).

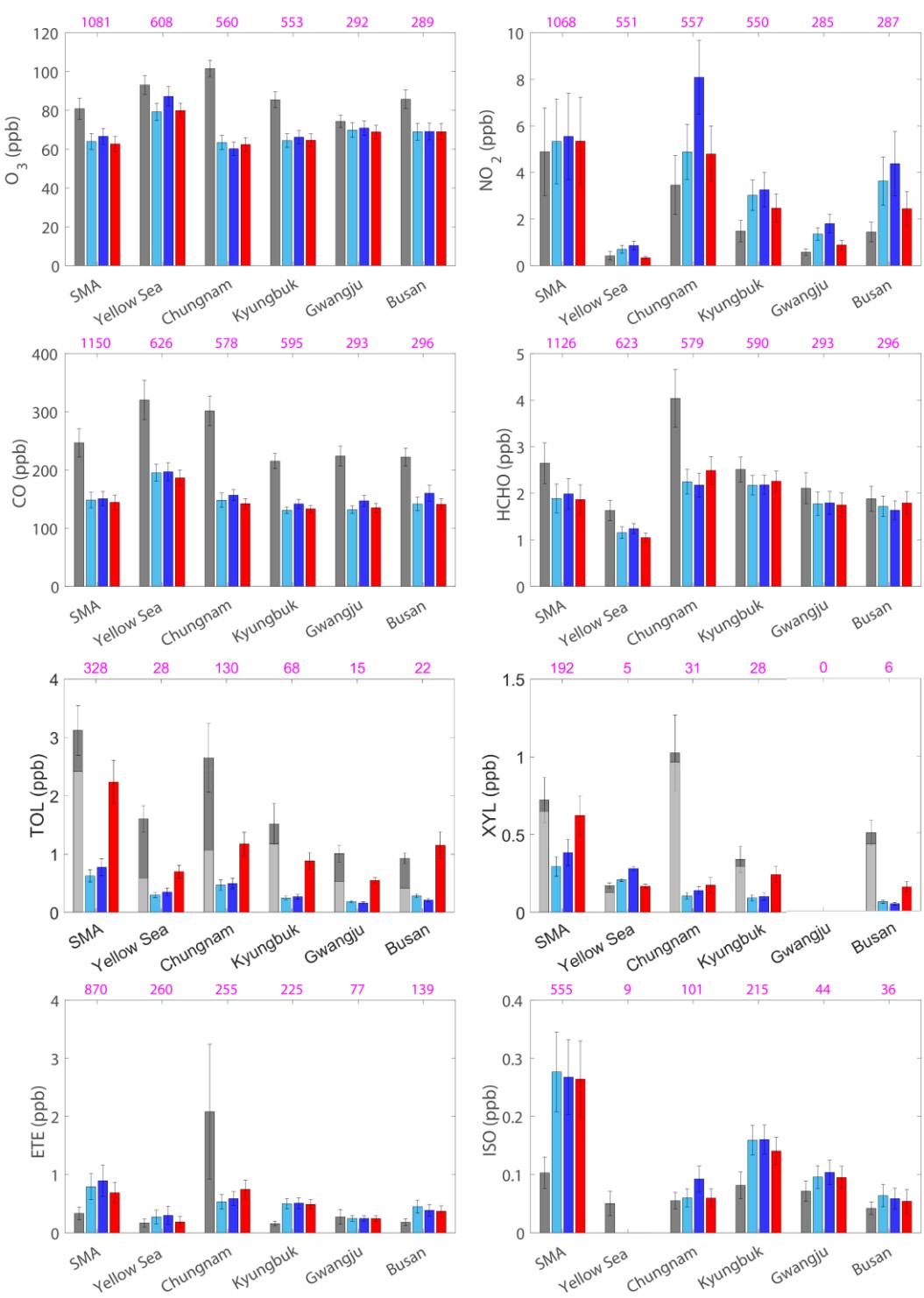

**Figure 8**. The mean (bars) and 1/4 of standard deviations (whiskers) of (a) $O_3$, (b) $NO_2$, (c) CO, (d) HCHO, (e) TOL, (f) XYL, (g) ethene (ETE), and (h) isoprene (ISO) (unit = ppb) from DC-8 (dark grey), EDV2 (sky blue), EDV3 (blue), and KOV5 (red) for each box are shown, respectively. TOL and XYL are calculated based on Table S8 (Supporting Information). The contribution of toluene to TOL and m/p-Xylene + o-Xylene to XYL is represented with light grey bars (e, f). The sampling numbers are represented with magenta color above the plots.

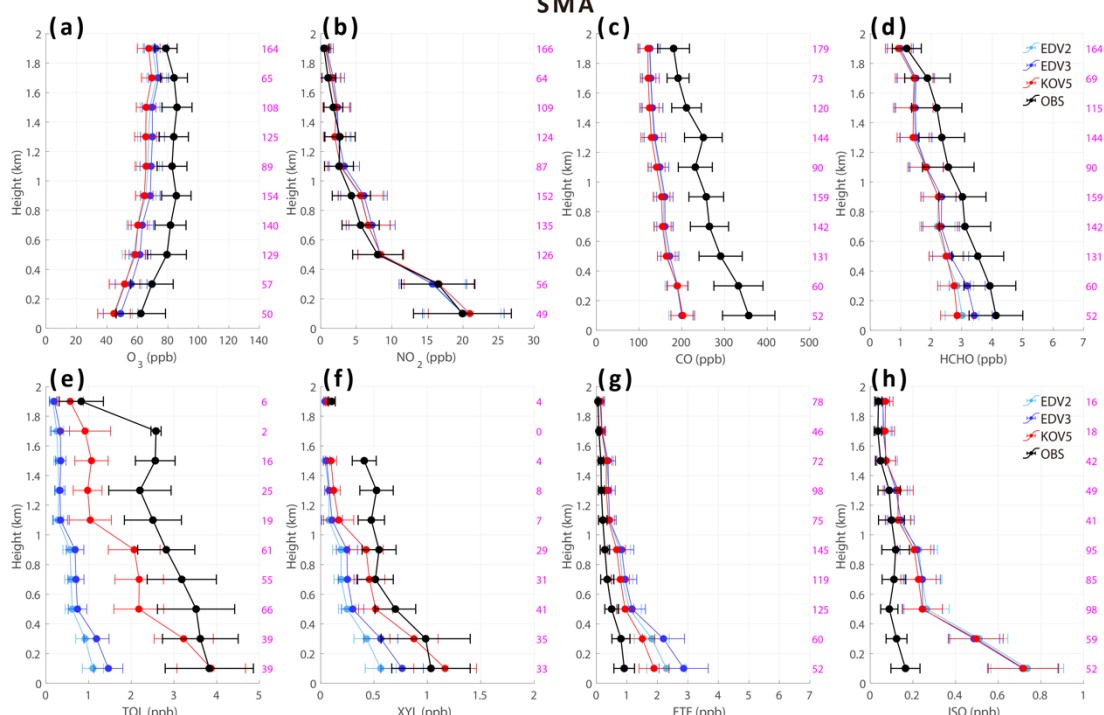

Figure 9. Vertically averaged (a) $O_3$, (b) $NO_2$, (c) CO, (d) HCHO, (e) TOL, (f) XYL, (g) ETE, and (h) ISO from DC-8 (black), EDV2 (sky blue), EDV3 (blue), and KOV5 (red) in SMA under 2 km height above ground level. The 1/2 of standard deviations are represented with black whiskers in each 200m layer. The sample number is presented with magenta color on the right side of the plots.

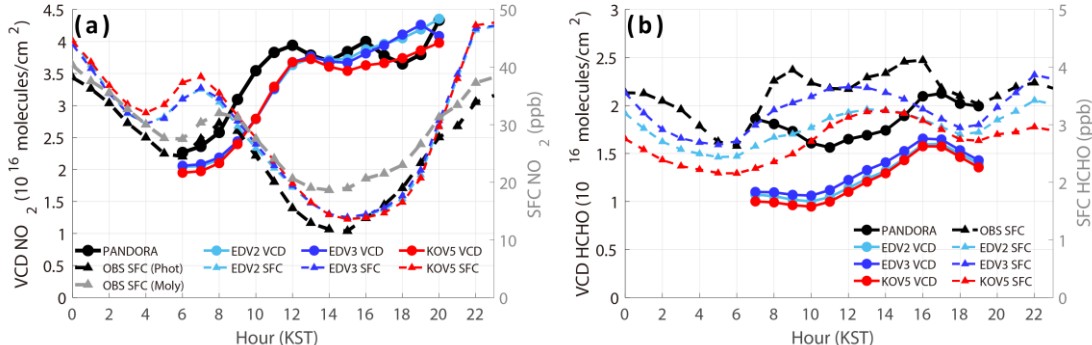

**Figure 10.** The diurnal cycles of vertical columns and surface concentrations of (a) $NO_2$
and (b) HCHO from Pandora spectrometer (column), and ground-based instruments
(TEI 42i $NO_x$ analyzer and Aerodyne QCL) at the Olympic Park site (37.5232˚N,
127.126˚E). Surface concentrations of $NO_2$ are obtained by the two methods:
molybdenum converter and photolytic method. EDV2 (sky blue), EDV3 (blue), and
KOV5 (red) are compared with observations. The WRF-Chem vertical column
concentrations are produced by summing all vertical layers.

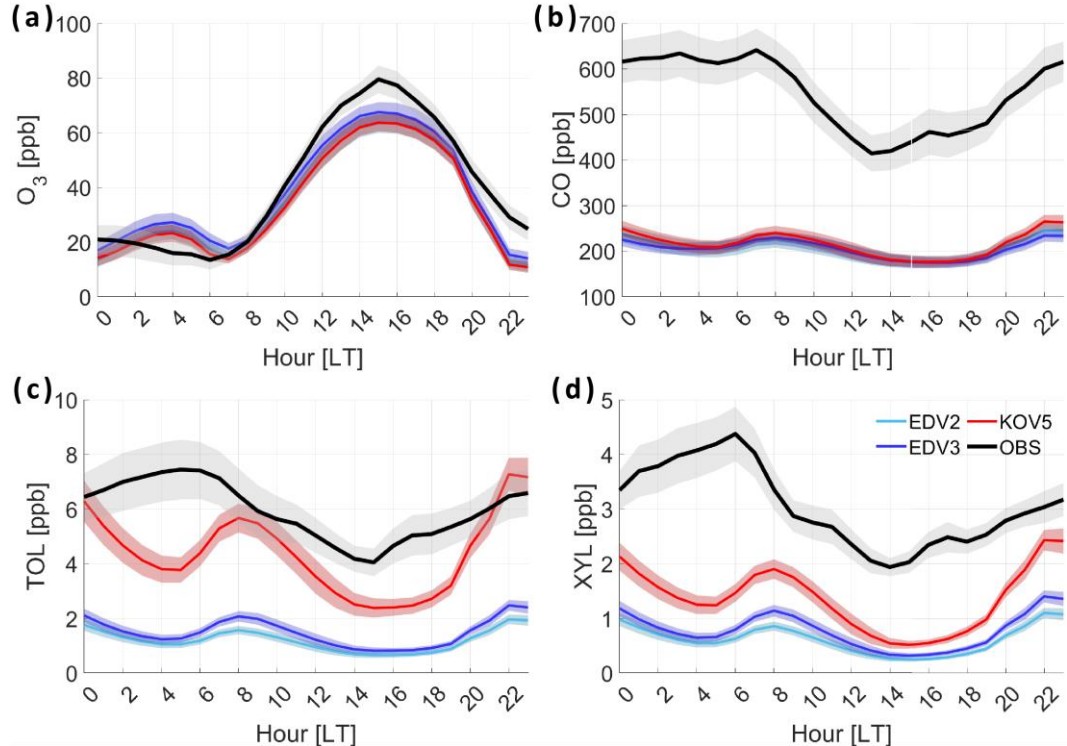

**Figure 11.** Diurnal cycles of surface (a) $O_3$, (b) CO, (c) TOL, and (d) XYL at the Olympic Park site. EDV2 (sky blue), EDV3 (blue), and KOV5 (red) are compared with the observations. 1/4 of standard deviations are represented with grey shades. The average period is from the 11th May to the 10th June.

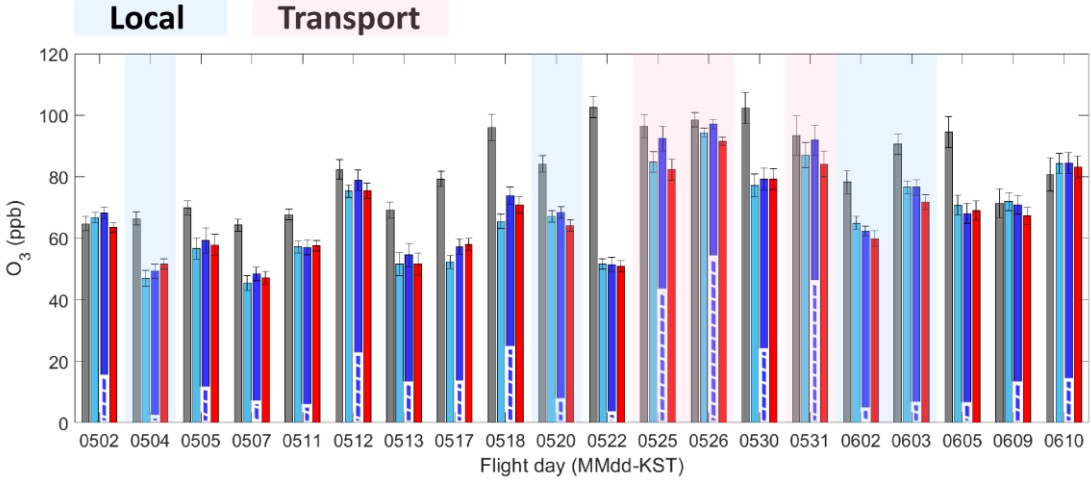

**Figure 12.** Averaged O₃ (bars) and 1/4 of standard deviations (whiskers) (unit: ppbv)
for the 20 DC8 flights (under 2 km height). The observations (grey) are compared with
the model results utilizing EDV2 (sky blue), EDV3 (blue), and KOV5 (red). White
hatch-filled bars over blue bars are the contribution of Chinese emissions to O₃
concentrations obtained from the default and sensitivity model runs with/without
Chinese anthropogenic emissions. The Local (5/4,20 and 6/2,3) and Transport
(5/25,26,31) cases are shaded with light blue and orange, respectively.

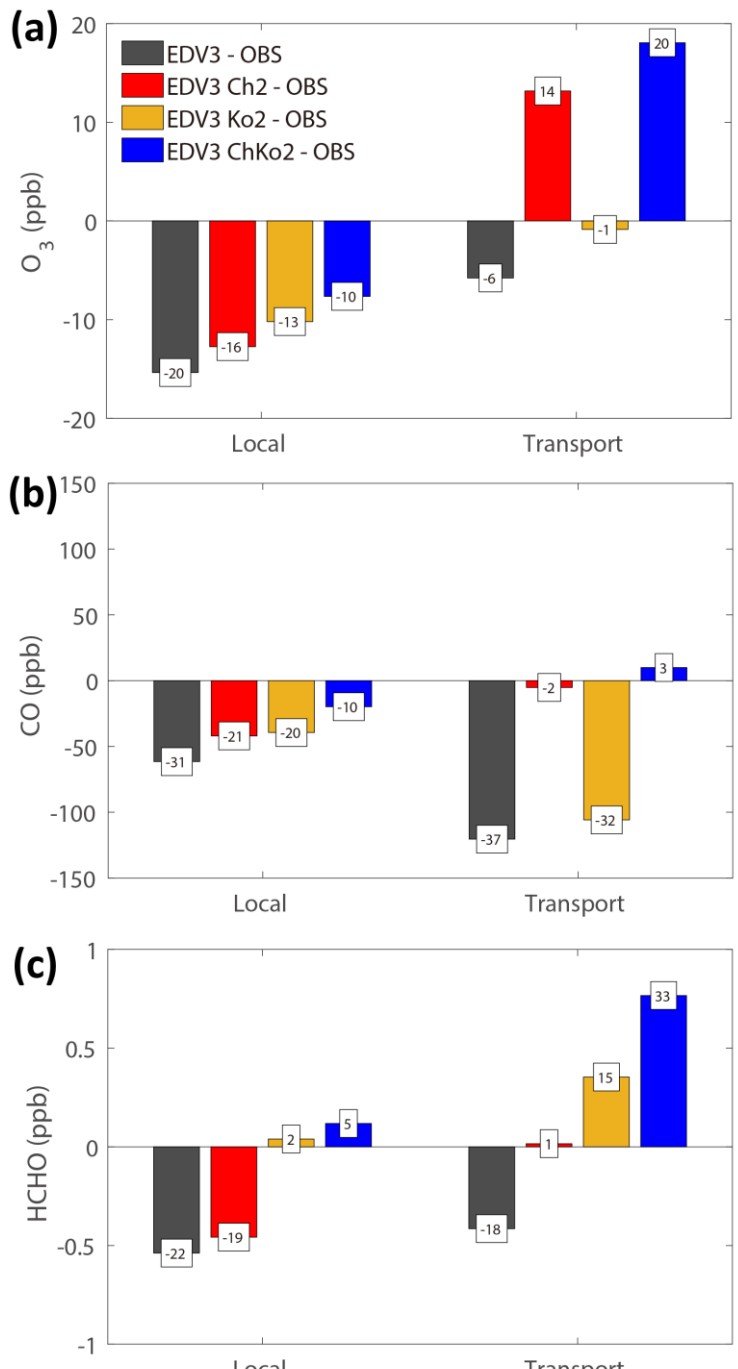

**Figure 13.** The biases in (a) the model O$_3$, (b) CO, and (c) HCHO concentrations (bars)
relative to the DC-8 observations under 2 km height over SMA (dark gray: EDV3, red:
EDV3 Ch2, orange: EDV3 Ko2, red: EDV3_ChKo2): (left panel) Local and (right panel)
Transport case. Fractional differences (%) are shown in the white boxes.

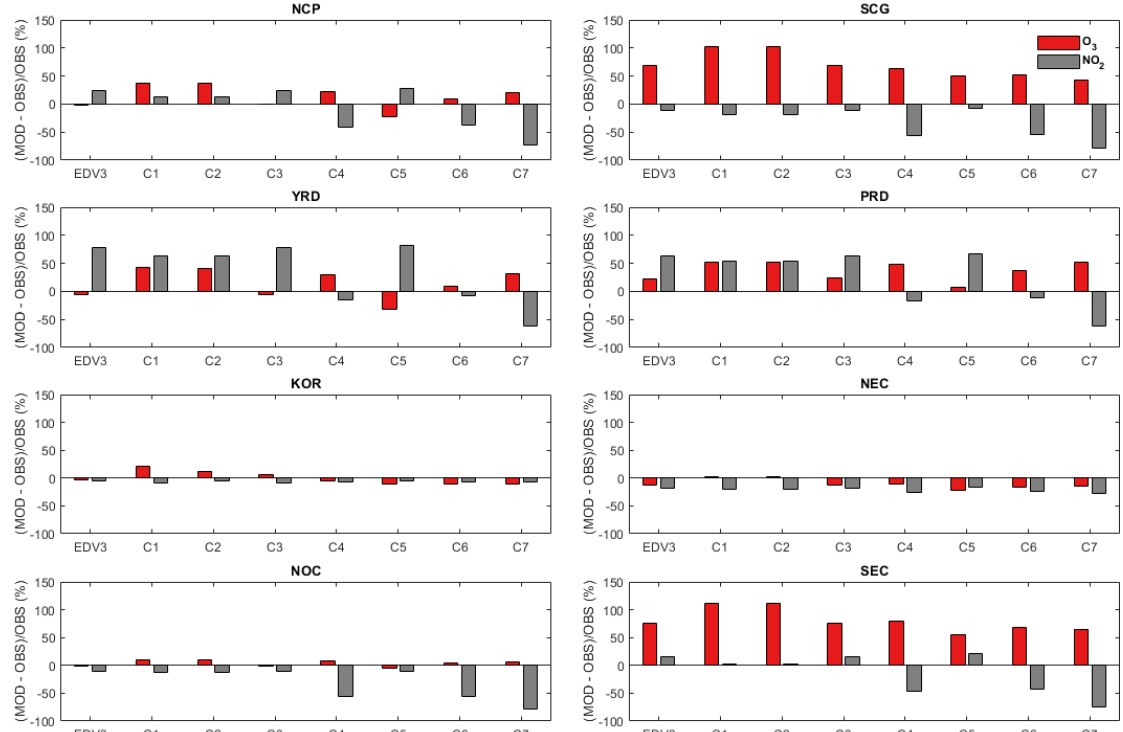

**Figure 14**. Comparison of relative biases ((Model-Observation)/Observation, unit=%) of daily $O_3$ and $NO_2$ at surface observation sites during the KORUS-AQ campaign period from sensitivity simulation (C1-7) with EDV3 in each region (NCP, SCG, YRD, PRD, KOR, NEC, NOC, and SEC). C1; EDGAR-HTAP v3 with double CO and VOC emission in China and South Korea, C2; EDGAR-HTAP v3 with double CO and VOC emission in China, C3; EDGAR-HTAP v3 with double CO and VOC emission in South Korea, C4; EDGAR-HTAP v3 with 50% NOx reduction in China, C5; EDGAR-HTAP v3 with 50% VOC reduction in China, C6; EDGAR-HTAP v3 with 50% NOx and VOC reduction in China, C7; EDGAR-HTAP v3 with 75% NOx reduction in China.

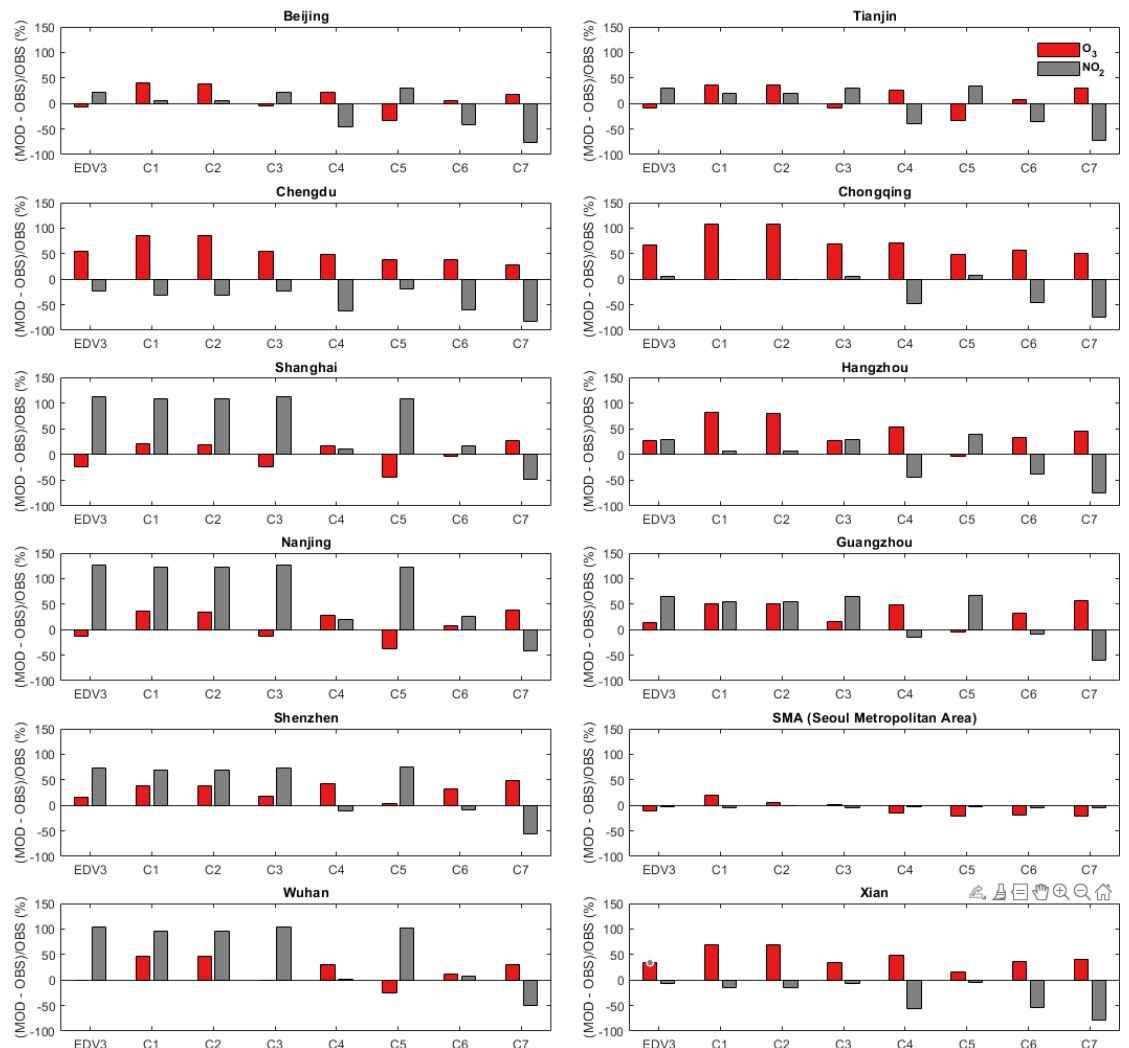

Figure 15. Same as Figure 14 except that the region is changed to cities; Beijing (39.4-41.1N, 115.4-117.5E), Tianjin (38.55-40.25N, 116.7-118.1E), Chengdu (30.05-31.5N, 103-105E), Chongqing (28.15-32.25N, 105.3-110.2E), Shanghai (30.7-31.5N, 120.85-122E), Hangzhou (29.2-30.6N, 118.3-120.9E), Nanjing (31.2-32.65N, 118.35-119.25E), Guangzhou (22.55-24N, 112.9-114.05E), Shenzhen (22.4-22.9N, 113.7-114.65E), SMA (37.2-37.8N, 126.5-127.3E), Wuhan (29.95-31.4N, 113.65-115.1E), and Xian (33.65-34.75N, 107.65-109.9E).