# Peer review of "Sensitivity of the WRF-Chem v4.4 ozone, formaldehyde, and their precursors"

_Geoscientific Model Development, 2023_

## Author Comment (AC1)

**Response to Reviewer 1's Comments**

The reviewer's comments are written in blue and our responses are in black.

The authors ran WRF-Chem simulations over East Asia using three different emission inventories (EDGAR-HTAP v2, EDGAR-HTAP v3, and KORUS v5) and compared the model output to three sets of observations (routine monitoring data, airborne KORUS-AQ data, and ground-based KORUS-AQ data). They also ran sensitivity tests to doubling CO and VOC emissions and probed how the chemistry changed. Such comparisons are useful for model development, but I think there is opportunity for the paper to be strengthened in the following ways:

1. Currently, $O_3$ and $NO_2$ are treated separately in the comparison. I suggest adding a comparison of odd-oxygen ($O_x = O_3 + NO_2$) to probe whether the model issue is too much $O_3$ titration by NO or problems with the $O_3$ production regime.

→ We added analysis of Ox with surface observations in China and South Korea after P14 L24 (also see **Figure R1 and R2**). The diurnal patterns of Ox are well simulated with all emission inventories (**Figure R1**), showing similar issues that are previously discussed in section 3.2.

As the reviewer expected, underestimations in the model $O_3$ in YRD and NCP using EDV2 (light blue lines) disappeared when it was replaced by $O_x$, suggesting that the $O_3$ biases using EDV2 in the regions are caused by too much $NO_x$ titration or inefficient $O_3$ formation in a NOx-saturated regime. For other regions and cases using EDV3 and KOV5, Ox plots highlight biases in the model $NO_2$ levels. In YRD, Ox overestimations correspond to $NO_2$ overestimations in Figure 4. Meanwhile, in SCG and SEC, there are Ox overestimations caused by $O_3$ overestimations, suggesting a potential VOC emission overestimation.

Detailed descriptions, along with Figure R1 and R2, are included in the revised manuscript and Supporting Information.

[Figure]

**Figure R1**. Averaged Ox concentrations from ground-based observations and model simulations over the areas that distinguish urban (red box) and non-urban (green box) region (central plot). Box-averaged diurnal cycle (solid lines) of Ox and 1/4 of standard deviations (filled area) from observations (black), EDV2 (sky blue), EDV3 (blue), and KOV5 (red) by local time are shown.

[Figure]

**Figure R2**. Comparison of (a) the campaign averaged ground-based maximum daily average of 8-hour Ox (MDA8 Ox) (unit: ppb) observations and WRF-Chem simulations with (d) EDGAR-HTAP v2 (EDV2), (e) v3 (EDV3), (f) KORUS v5 (KOV5) and (g, h, i) the differences between the observations and model results. The scatter plots comparing averaged observations and the three-emission-based WRF-Chem simulations (sky blue; EDV2, blue; EDV3, red; KOV5) are shown in (b) and (c) for Eastern China and South Korea, respectively.

2. On a related note, can you use the individual comparisons you've done of VOCs, NOx, and $O_3$ in different regions with each inventory to draw some conclusions about how biases in either or both $NO_x$ or VOC emissions affect $O_3$ predictions / chemical regimes in the different regions? There is a little bit of this on pages 13-14, but more organized conclusions about this (especially with your sensitivity tests to doubling the CO and VOC emissions) would be very useful for future model interpretation and emissions inventory development. For example, on page 12, can you discuss why the differences in VOCs and $NO_x$ in each of the inventories cause them to simulate $O_3$ differently?

→ We provided **Table R1**, detailing NOx, TOL, XYL, biogenic isoprene emissions, and formaldehyde-to-NO$_2$ ratio (FNR) for each region and emission inventory to enhance the understanding of regional differences. We included some discussions in section 3.2, such as the descriptions of VOC-limited regime in NCP with low FNR (< 1). The higher emissions of TOL and XYL in EDV3 and KOV5 resulted in higher O$_3$ concentrations with the smaller biases than EDV2 (**Table R1**). In SCG and SEC, biogenic emissions exceeded TOL and XYL by up to the factor of 10 with all emission inventories.

**Table R1**. Comparison of total NOx, TOL, XYL, biogenic isoprene emissions in May, and formaldehyde-to-NO$_2$ ratio (FNR) for the KORUS-AQ campaign period for different emission datasets in each regional box. The MEGAN biogenic isoprene emissions are equally applied to all simulations using different emission data. (unit = mol/s for emissions)

| Type | emissions | NCP | SCG | YRD | PRD | KOR(SMA) | NEC | NOC | SEC |
|---|---|---|---|---|---|---|---|---|---|
| **NOx emission** | **EDV2** | 5967 | 1500 | 2366 | 1178 | 990(196) | 987 | 688 | 590 |
| | **EDV3** | 5202 | 1654 | 1642 | 1091 | 1191(214) | 876 | 597 | 662 |
| | **KOV5** | 3237 | 902 | 1166 | 607 | 886(191) | 513 | 373 | 410 |
| **TOL emission** | **EDV2** | 140 | 56 | 84 | 47 | 27(6) | 26 | 8 | 20 |
| | **EDV3** | 220 | 77 | 99 | 68 | 27(8) | 40 | 9 | 36 |
| | **KOV5** | 403 | 106 | 234 | 155 | 98(26) | 68 | 21 | 79 |
| **XYL emission** | **EDV2** | 84 | 34 | 51 | 28 | 15(4) | 15 | 4 | 12 |
| | **EDV3** | 132 | 46 | 60 | 41 | 16(4) | 24 | 6 | 22 |
| | **KOV5** | 133 | 35 | 79 | 52 | 41(9) | 21 | 7 | 26 |
| **Biogenic isoprene emission** | | 132 | 364 | 43 | 127 | 135(6) | 106 | 23 | 310 |
| **FNR (14-16LT)** | **EDV2** | 0.25 | 1.31 | 0.19 | 0.52 | 0.53(0.19) | 0.68 | 0.76 | 1.18 |
| | **EDV3** | 0.44 | 1.30 | 0.32 | 0.52 | 0.43(0.18) | 0.93 | 0.94 | 1.33 |
| | **KOV5** | 0.72 | 2.33 | 0.48 | 1.00 | 0.71(0.22) | 1.44 | 1.49 | 1.91 |

Interpreting O$_3$ biases using FNR is cautioned due to the complex interplay of VOC and NOx emissions and chemistry. Therefore, we added section 4 (discussion) with 4 additional sensitivity simulations (C5-C8 in **Figure R3**) as discussed in Kim et al. (2023), providing insights for O$_3$ bias correction in each region and city.

  In SCG and SEC, the C5 case (50% anthropogenic VOC emission reduction only) exhibited the lowest O$_3$ biases, with a slight decrease in O$_3$ concentrations in the C4 case (50% NOx reduction only), implying the need to reduce VOC emissions (biogenic and/or anthropogenic

emissions) (**Figure R3**). For the YRD and PRD, both NOx and VOC emissions should be reduced based on C6 case (50% NOx and VOC reduction), while C4 case (only NOx 50% reduction) increased $O_3$ bias.

We also compared the sensitivity simulations with 12 mega cities in China and South Korea (**Figure R4**). VOC 50% reduction (C5 case) improved $O_3$ and $NO_2$ simulations in Chengdu and Chongqing. The lowest biases of $O_3$ and $NO_2$ were achieved with 50% NOx and VOC reduction case (C6 case) for Shanghai, Nanjing, Guangzhou, Shenzhen, and Wuhan.

The detailed analysis, along with Figure R3 and R4, has been added to the revised manuscript.

[Figure]

**Figure R3**. Comparison of relative biases ((Model-Observation)/Observation, unit=%) of daily $O_3$ and $NO_2$ at surface observation sites during the KORUS-AQ campaign period from sensitivity simulation (C1-7) with EDV3 in each region (NCP, SCG, YRD, PRD, KOR, NEC, NOC, and SEC). C1; EDGAR-HTAP v3 with double CO and VOC emission in China and South Korea, C2; EDGAR-HTAP v3 with double CO and VOC emission in China, C3; EDGAR-HTAP v3 with double CO and VOC emission in South Korea, C4; EDGAR-HTAP v3 with 50% NOx reduction in China, C5; EDGAR-HTAP v3 with 50% VOC reduction in China, C6; EDGAR-HTAP v3 with 50% NOx and VOC reduction in China, C7; EDGAR-HTAP v3 with 75% NOx reduction in China.

[Figure]

**Figure R4**. Same as **Figure R3** except that the region is changed to cities; Beijing (39.4-41.1N, 115.4-117.5E), Tianjin (38.55-40.25N, 116.7-118.1E), Chengdu (30.05-31.5N, 103-105E), Chongqing (28.15-32.25N, 105.3-110.2E), Shanghai (30.7-31.5N, 120.85-122E), Hangzhou (29.2-30.6N, 118.3-120.9E), Nanjing (31.2-32.65N, 118.35-119.25E), Guangzhou (22.55-24N, 112.9-114.05E), Shenzhen (22.4-22.9N, 113.7-114.65E), SMA (37.2-37.8N, 126.5-127.3E), Wuhan (29.95-31.4N, 113.65-115.1E), and Xian (33.65-34.75N, 107.65-109.9E).

3. Most of the conclusions in the manuscript are stated as "X is biased low with Y emissions inventory." These statements would be more useful to the atmospheric chemistry community writ large if those statements were extended to say, "X is biased low with Y emissions inventory, which has Z implications for our understanding of emissions/chemistry." For example, on page 20 line 12, can you add something to this sentence about the implications of having larger biases in the Transport case compared to the Local case? For a second example, on page 20, line 20, can you add something about the implications for $NO_x$ emissions (based on $O_3$ being wrong but CO and HCHO being largely okay)? There are many other instances in the manuscript where this would be useful, but hopefully, the two examples I provided here are helpful illustrations.

→ To enhance discussions about causes of the model $O_3$ biases, we added a separate section of discussion about the chemical regimes in each region and city and the best way to reduce ozone biases accordingly, incorporating NOx emissions information. Please refer to our response to Reviewer's major comment 2. Furthermore, we added the sentences about analysis of Local and Transport case.

The excessive $O_3$ with double emissions in China is attributed to an overestimation of background $O_3$. We included Figure R5 to represent the overestimated $O_3$ from the downwind area (Yellow Sea) when CO and VOC emissions are doubled in China. This analysis is included in the revised manuscript. Furthermore, in section 4, causes of $O_3$ biases and directions to improvement for each region and cities are suggested in detail including Figure R3 and R4.

[Figure]

**Figure R5**. Vertically averaged $O_3$ from DC-8 (black), EDV2 (sky blue), EDV3 (blue), KOV5 (red), EDV3 with doubling Chinese CO and VOC emissions (dashed blue), EDV3 with doubling Korean CO and VOC emissions (dotted blue), and EDV3 with doubling Chinese and Korean CO and VOC emissions (dotted dashed blue) in Yellow Sea under 2 km height above ground level. The 1/2 of standard deviations are represented with whiskers in each 200m layer. The sample number is presented with magenta color on the right side of the plots.

4. The manuscript includes some contextualization of this work in the context of other emissions inventory comparisons (e.g., for CO on page 8). I think the paper would be strengthened by adding similar contextualization for the other comparisons ($NO_x$, $O_3$, VOCs, etc.), especially given how many model-measurement comparisons have been done to date with KORUS-AQ (and related) data.

→ Our objective is to systematically identify and summarize potential issue associated with anthropogenic bottom-up emission inventories, investigating their potential impact on $O_3$

simulations in East Asia. We included relevant previous model studies in section 1 (Introduction) as explained below.

"Many modeling studies are done during this period including validations of CTM results with various observations. Miyazaki et al. (2019) adjusted emission inventories using various satellite data sets and Model for Interdisciplinary Research on Climate with chemistry (MIROC-Chem) resulting in $O_3$ simulations improvement. Goldberg et al. (2019) reported underestimations of NOx emissions in South Korea including Seoul. Souri et al. (2020) also revealed the same issue in South Korea and analyzed sensitivity of $O_3$ formation to the NOx and VOC emission adjustments derived from inverse modeling. Tang et al. (2019) revealed negative bias of simulated CO concentrations in East Asia by utilizing satellite data and the Community Atmosphere Model with Chemistry (CAM-Chem). Choi et al. (2022) modified anthropogenic VOC emissions using satellite HCHO observations and inverse modeling method with the Goddard Earth Observing System with Chemistry (GEOS-Chem), which reduced $O_3$ and HCHO biases."

**Specific comments**:

Page 5, line 8: are the NMVOCs lumped or speciated?

→ It is lumped NMVOC. We added 'total' in front of 'non-methane volatile organic compound'.

Page 5, line 9: do you apply any scale factors for using 2010 emissions data in a 2016 simulation?

→ We did not use scale factors.

Page 5, line 17: What does 'specifically' mean here?

→ We intended to describe that it is speciated NMVOCs from EDGAR-HTAP v2. We will change 'specifically mapped EDGAR-HTAP v2 data' to 'speciated EDGAR-HTAP v2 VOC data' to avoid confusion.

Page 7, lines 1-2: should read "…toluene and less *reactive* aromatics…"

→ Thank you. We added 'reactive' in front of 'aromatics' in the revised manuscript.

Page 7, line 14: what species are you referring to that is larger in South Korea by 263%?

→ It's TOL as mentioned in previous sentence. We added 'of TOL' behind 'relative difference' to avoid confusion in the revised manuscript.

Page 8, line 1: add "respectively" after "(HCHO)"

→ Thank you for the comment. We included 'respectively' after '(HCHO)'.

Page 8, line 13: I think it would be clearer to say "For all emission inventories…" rather than "With all emission…"

→ Thank you for the comment. We changed "With all emission inventories" to "For all emission inventories" in the updated version of manuscript.

Page 8, lines 14-17: It was hard for me to figure out which simulations correspond to which numbers in these sentences. Reword to clarify?

→ We changed the sentence to "we conducted two additional model simulations using EDGAR-HTAP v3 that shows lowest bias of O3 concentrations compared to DC-8 than

EDGAR-HTAP v2 and KORUS v5 over the SMA (mean bias = EDV2: -16.9, EDV3: -14.2, KOV5: -18.1 ppb)" adding bias information between parentheses in the revised manuscript.

Page 10, line 19: What kind of interpolation method was used?

→ The linear interpolation method is used for the vertical interpolation. We added "using linear interpolation method" after "vertically interpolated to the aircraft data" in the revised manuscript.

Page 11, lines 11-19: Can you say something about how much these meteorological biases affect your comparisons? For example, how much would temperature-dependent evaporative VOC emissions change over these temperature ranges?

→ Both anthropogenic and biogenic VOC emissions can be affected by air temperature (Huang et al., 2022; Song et al., 2019). In this response, we could calculate the impact of temperature on biogenic VOC emissions.  The isoprene emission in MEGAN is calculated following the equation below (Guenther et al., 2006).

$$Emission = [EF][\gamma][\rho]$$

$$\gamma_{Temp} = E_{opt} \cdot \frac{C_2 \cdot \exp{(C_1 \cdot x)}}{C_2 - C_1 \cdot (1 - \exp{(C_2 \cdot x)})}$$

$$x = \frac{\left(\frac{1}{T_{opt}} - \frac{1}{T}\right)}{0.00831}$$

EF is emission factor (mg m$^{-2}$ h$^{-1}$). $\rho$ is normalized ratio. $\gamma$ is an emission activity factor that can vary for different conditions such as leaf area index, temperature, vegetation type, leaf age, soil moisture, and canopy environment. $E_{opt}$ and $T_{opt}$ are empirical coefficients. $C_1$ and $C_2$ are constants. We calculated isoprene (ISO) emission sensitivity to temperature bias at each SYNOP station by changing T. The negative temperature biases resulted in reduced

isoprene emissions in South Korea (**Figure R6**). However, as discussed in 3.3.1, ISO is still overestimated for all regions.

[Figure]

**Figure R6**. Relative isoprene (ISO) emission change from the temperature bias at the surface (unit = %).

Page 12, line 11: "all emissions inventories" instead of "all emissions"

→ Thank you for the comment. We changed "all emissions" to "all emission inventories".

Page 17, line 3: ISO definition should be moved earlier to where it's first used.

→ Agreed. We first defined isoprene as ISO in line 24 of page 13 in the revised manuscript.

Page 19, lines 23-24: can you use the biases calculated during the local case to draw some conclusions about the emissions inventory over China?

→ We added ", which implies that the insufficient local emissions of $O_3$ precursors in the emission inventories are much important that the Chinese emissions." after "15.5-18.2 ppb" in the revised manuscript to clarify local VOC emission issues to the low model $O_3$ concentrations in South Korea.

Title: Unclear what 'precursor' refers to here. Is it $O_3$ and HCHO precursors? If so, perhaps rephrasing it as "ozone, formaldehyde, and their precursors" would be clearer.

→ Agreed. We changed the title.

**References**

Choi, J., Henze, D. K., Cao, H., Nowlan, C. R., Abad, G. G., Kwon, H.-A., Lee, H.-M., Oak, Y. J., Park, R. J., Bates, K. H., Massakkers, J. D., Wisthaler, A., and Weinheimer, A. J.: An Inversion Framework for Optimizing Non-Methane VOC Emissions Using Remote Sensing and Airborne Observations in Northeast Asia During the KORUS-AQ Field Campaign, *J. Geophys. Res. Atmos.*, 127, e2021JD035844, https://doi.org/10.1029/2021JD035844, 2022.

Kim, S.-W., Kim, K.-M., Jeong, Y., Seo, S., Park, Y., and Kim J.: Changed in surface ozone in South Korea on diurnal to decadal timescales for the period of 2001-2021, *Atmos. Chem. Phys.*, 23, 12867-12886, https://doi.org/10.5194/acp-23-12867-2023, 2023.

Goldberg, D. L., Saide, P. E., Lamsal, L. N., de Foy, B., Lu, Z., Woo, J.-H., Kim, Y., Kim, J., Gao, M., Carmichael, G., and Streets, D. G.: A top-down assessment using OMI NO2 suggests an underestimate in the NOx emissions inventory in Seoul, South Korea, during KORUS-AQ, *Atmos. Chem. Phys.*, 19, 1801-1818, https://doi.org/10.5194/acp-19-1801-2019, 2019.

Guenther, A., Karl, T., Harley, P., Wiedinmyer, C., Palmer, P. I., and Geron, C.: Estimates of global terrestrial isoprene emissions using MEGAN (Model of Emissions of Gases and Aerosols from Nature), *Atmos. Chem. Phys.*, 6, 3181-3210, https://doi.org/10.5194/acp-6-3181-2006, 2006.

Huang, J., Yuan, Z., Duan, Y., Liu, D., Fu, Q., Liang, G., Li, F., and Huang, X.: Quantification of temperature dependence of vehicle evaporative volatile organic compound emissions from different fuel types in China, *Sci. Total Environ.*, 813, 152661, https://doi.org/10.1016/j.scitotenv.2021.152661, 2022.

Miyazaki, K., Sekiya, T., Fu, D., Bowman, K. W., Kulawik, S. S., Sudo, K., Walker, T., Kanaya, Y., Takigawa, M., Ogochi, K., Eskes, H., Boersma, K. F., Thompson, A. M., Gaubert, B., Barre, J., and Emmons, L. K.: Balance of Emission and Dynamical Controls on Ozone During the Korea-United States Air Quality Campaign From Multiconstituent Satellite

Data Assimilation, *J. Geophys. Res. Atmos.*, 124, 387-413, https://doi.org/10.1029/2018JD028912 , 2019.

Song, C., Liu, B., Dai, Q., Li, H., and Mao, H.: Temperature dependence and source apportionment of volatile organic compounds (VOCs) at an urban site on the north China plain, *Atmos. Environ.*, 207, 167-81, https://doi.org/10.1016/j.atmosenv.2019.03.030, 2019.

Souri, A. H., Nowlan, C. R., Abad, G. G., Zhu, L., Blake, D. R., Fried, A., Weinheimer, A. J., Wisthaler, A., Woo, J.-H., Zhang, Q., Chan Miller, C. E., Liu, X., and Chance, K.: An inversion of NOx and non-methane volatile organic compound (NMVOC) emissions using satellite observations during the KORUS-AQ campaign and implications for surface ozone over East Asia, *Atmos. Chem. Phys.*, 20, 9837-9854, https://doi.org/10.5194/acp-20-9837-2020, 2020.

Tang, W., Emmons, L. K., Arellano Jr, A. F., Gaubert, B., Knote, C., Tilmes, S., Buchholz, R. R., Pfister, G. G., Diskin, G. S., Blake, D. R., Blake, N. J., Meinardi, S., DiGangi, J. P., Choi, Y., Woo, J.-H., He, C., Schroeder, J. R., Suh, I., Lee, H.-J., Kanaya, Y., Jung, J., Lee, Y., and Kim, D.: Source Contributions to Carbon Monoxide Concentrations During KORUS-AQ Based on CAM-chem Model Applications, *J. Geophys. Res. Atmos.*, 124, 2796-2822, https://doi.org/10.1029/2018JD029151, 2019.

---

## Author Comment (AC2)

Responses to Reviewer 2's comments

The reviewer's comments are written in blue and our responses are in black.

The authors provide an overview of emission inventories and model simulations in East Asia using WRF-Chem v4.4 with three different emission inventories. To do so, they compare model results with various observations and conduct a sensitivity test by doubling CO and VOC emissions in China and South Korea. This study is significant as it shows the current state of emission inventories and represents points to improve simulated ozone. However, the following comments should be considered to enhance the study.

General comments:

1. In my opinion, the authors need to clarify the main topic and purpose of this study. I think that the research aims to evaluate model simulations using three emission inventories, comparing them with various observations and to conduct an analysis and sensitivity test of underestimated ozone. However, these aspects do not seem to be adequately explained in the introduction and abstract. Additionally, it should be made clear whether the analysis of underestimated ozone will focus on East Asia or be specific to South Korea. If the authors want to focus on East Asia, additional analysis for China has to be included, as discussed for South Korea. The scope of regions needs to be clarified.

→ To emphasize the focus and purpose of this paper, we clarified our objectives in the introduction and abstract. Previous modeling studies during the KORUS-AQ campaign period utilized old version of anthropogenic bottom-up emission inventories, using inversion or data assimilation method for the accurate air pollutants simulations ($O_3$, CO, HCHO, etc.). In contrast, our study employed recent versions of bottom-up emission datasets (EDGAR-HTAP v3 and KORUS v5) and expanded the analysis domain from South Korea to East Asia, including China. The revised manuscript includes brief descriptions of previous works (Choi et al., 2022; Goldberg et al., 2019; Miyazaki et al., 2019; Souri et al., 2020; Tang et al., 2019) are included in the introduction, clarifying our focus on $O_3$ and its precursors in both the abstract and introduction.

2. The paper discusses which emissions should improve and what causes the underestimation of ozone, comparing simulated species with observations. However, the discussion needs to be organized more. It mainly focuses on the underestimation of VOC emissions about underestimated ozone, but the impact of VOC emissions can vary depending on the ozone production regime (relative ratio to NOx emissions). The paper independently compares VOC, NOx, and $O_3$ mixing ratios, but these species are related to each other, and more effort considering them together is needed to understand model performance and emissions. Therefore, additional explanations about these regimes are necessary for the analysis during KORUS-AQ. In addition, NOx emissions could contribute to VOC chemistry, resulting in ozone changes, so it would be great to describe them in the sensitivity analysis.

$\rightarrow$ We agree with the suggestion that the simulated $O_3$ should be interpreted along with $NO_2$ and VOC emissions. We included **Table R1** in the revised manuscript to fully represent the differences between emission inventories and their simulated chemical regimes in different regions. Also, we explained the NOx, VOC, biogenic isoprene emissions, and formaldehyde-to-$NO_2$ ratio in each region.

Specifically, in NCP, as the model simulated NCP as NOx-saturated regime (FNR < 1), KOV5 simulated $O_3$ concentrations well because of higher reactive VOC emissions (TOL and XYL). In SCG and SEC, relatively high biogenic emissions from MEGAN compared to TOL and XYL led to high FNR (FNR > 1). These interpretations are incorporated into section 3.2.

**Table R1**. Comparison of total NOx, TOL, XYL, biogenic isoprene emissions, and formaldehyde-to-NO$_2$ ratio (FNR) for different emission datasets in each regional box. The MEGAN biogenic isoprene emissions are equally applied to all simulations using different emission data. (unit = mol/s for emissions)

| Type | emissions | NCP | SCG | YRD | PRD | KOR(SMA) | NEC | NOC | SEC |
|---|---|---|---|---|---|---|---|---|---|
| **NOx emission** | EDV2 | 5967 | 1500 | 2366 | 1178 | 990(196) | 987 | 688 | 590 |
| | EDV3 | 5202 | 1654 | 1642 | 1091 | 1191(214) | 876 | 597 | 662 |
| | KOV5 | 3237 | 902 | 1166 | 607 | 886(191) | 513 | 373 | 410 |
| **TOL emission** | EDV2 | 140 | 56 | 84 | 47 | 27(6) | 26 | 8 | 20 |
| | EDV3 | 220 | 77 | 99 | 68 | 27(8) | 40 | 9 | 36 |
| | KOV5 | 403 | 106 | 234 | 155 | 98(26) | 68 | 21 | 79 |
| **XYL emission** | EDV2 | 84 | 34 | 51 | 28 | 15(4) | 15 | 4 | 12 |
| | EDV3 | 132 | 46 | 60 | 41 | 16(4) | 24 | 6 | 22 |
| | KOV5 | 133 | 35 | 79 | 52 | 41(9) | 21 | 7 | 26 |
| **Biogenic isoprene emission** | | 132 | 364 | 43 | 127 | 135(6) | 106 | 23 | 310 |
| **FNR (14-16LT)** | EDV2 | 0.25 | 1.31 | 0.19 | 0.52 | 0.53(0.19) | 0.68 | 0.76 | 1.18 |
| | EDV3 | 0.44 | 1.30 | 0.32 | 0.52 | 0.43(0.18) | 0.93 | 0.94 | 1.33 |
| | KOV5 | 0.72 | 2.33 | 0.48 | 1.00 | 0.71(0.22) | 1.44 | 1.49 | 1.91 |

To address the limitations of interpreting the efficient O$_3$ production regime with FNR, we additionally conducted sensitivity simulations with different emission; EDV3_Ch0.5NOx, EDV3_Ch0.5VOC, EDV3_Ch0.5NOxVOC, and EDV3_Ch0.25NOx, representing EDGAR-HTAP v3 with 50% NOx reduction, 50% VOC reduction, 50% NOx and VOC reduction, and 75% NOx reduction in China, respectively, as discussed in Kim et al. (2023).

Comparing relative biases of O$_3$ and NO$_2$ in each region and city (**Figure R1** and **R2**), the C5 case (50% VOC emission reduction only) exhibited the lowest O$_3$ biases in SCG and SEC, implying the need to reduce VOC emissions (biogenic and/or anthropogenic emissions) (**Figure R1**). In YRD and PRD, a 50% reduction in both NOx and VOC emissions (C6 case) produced the most reasonable O$_3$ and NO$_2$ simulations.

Additionally, we evaluated $O_3$ and $NO_2$ simulations with different emissions at 12 mega cities in China and South Korea (**Figure R2**). EDV3 simulated $O_3$ and $NO_2$ well for the cities such as Beijing, Tianjin, Hangzhou, SMA, and Xian. In Chengdu and Chongqing, high $O_3$ and $NO_2$ biases are alleviated with 50% VOC emission reduction. For Shanghai, Nanjing, Guangzhou, Shenzhen, and Wuhan, the C6 case shows the most reasonable $O_3$ and $NO_2$ simulations, while a simple 50% reduction of NOx slightly increased $O_3$ biases.

Those analysis are detailed in section 4 (discussion) in the revised manuscript.

[Figure]

**Figure R1**. Comparison of relative biases ((Model-Observation)/Observation, unit=%) of daily $O_3$ and $NO_2$ at surface observation sites during the KORUS-AQ campaign period from sensitivity simulation (C1-7) with EDV3 in each region (NCP, SCG, YRD, PRD, KOR, NEC, NOC, and SEC). C1; EDGAR-HTAP v3 with double CO and VOC emission in China and South Korea, C2; EDGAR-HTAP v3 with double CO and VOC emission in China, C3; EDGAR-HTAP v3 with double CO and VOC emission in South Korea, C4; EDGAR-HTAP v3 with 50% NOx reduction in China, C5; EDGAR-HTAP v3 with 50% VOC reduction in China, C6; EDGAR-HTAP v3 with 50% NOx and VOC reduction in China, C7; EDGAR-HTAP v3 with 75% NOx reduction in China.

[Figure]

**Figure R2**. Same as **Figure R1** except that the region is changed to cities; Beijing (39.4-41.1N, 115.4-117.5E), Tianjin (38.55-40.25N, 116.7-118.1E), Chengdu (30.05-31.5N, 103-105E), Chongqing (28.15-32.25N, 105.3-110.2E), Shanghai (30.7-31.5N, 120.85-122E), Hangzhou (29.2-30.6N, 118.3-120.9E), Nanjing (31.2-32.65N, 118.35-119.25E), Guangzhou (22.55-24N, 112.9-114.05E), Shenzhen (22.4-22.9N, 113.7-114.65E), SMA (37.2-37.8N, 126.5-127.3E), Wuhan (29.95-31.4N, 113.65-115.1E), and Xian (33.65-34.75N, 107.65-109.9E).

3. The font sizes in the figures are small, and the figure resolutions in both the manuscript and supplementary document are low. Please enhance their readability.

→ The resolution issue may be made when the file is converted to pdf file. It is fixed and all the figures are updated with the large font size and high resolution in the revised manuscript.

4. Many paragraphs are overly lengthy, attempting to cover multiple topics within a single paragraph. Please ensure that paragraphs are concise.

→ The paragraphs are organized based on the analysis region and species for clearer representation. The paragraphs are segmented in P3 L9, P9 L19, P11 L20, P12 L13, P12 L23, P16 L9, P16 L12, P17 L5, P17 L17, P18 L7, P18 L14, P20 L14, and P20 L20.

**Specific comments:**

P3 L3-15: In my opinion, the paragraph should be separated at L9. Additionally, before providing an overall description of the paper, it is important to clearly explain what the authors want to convey through the paper and what scientific significance it holds.

→ We added some references (Choi et al., 2022; Goldberg et al., 2019; Miyazaki et al., 2019; Souri et al., 2020; Tang et al., 2019;) that are previously conducted for emission adjustments using chemical transport model and added the descriptions of our purpose of this paper at P3 L9.

P4 L18: The link is not open.

→ As we checked again, this link is still working but direct connection to this site through PDF file is not working. We changed the link to "https://rda.ucar.edu/datasets/ds313.7/". We hope now this link is working.

P4 L21-22: Even so, in South Korea, the authors analyzed model performance over China. Therefore, the authors should discuss the effects of fire emissions on China.

→ We additionally simulated the WRF-Chem model using Fire Inventory from NCAR (FINN) v2.5 emissions (Wiedinmyer et al., 2022). The fire emission slightly increased averaged MDA8 $O_3$ concentrations by 1 ppbv (~ 1.6 %) in China. We added simple descriptions of this sensitivity test after "small impact on air quality simulations during the KORUS-AQ campaign period" at P4 L22 in the revised manuscript.

[Figure]

**Figure R3**. The (a) absolute and (b) relative differences of averaged MDA8 O₃ during the KORUS-AQ campaign period at the Chinese surface observations sites between WRF-Chem simulations with EDGAR-HTAP v3 (EDV3) and EDGAR-HTAP v3 with fire emissions (EDV3_Fire) (EDV3_Fire – EDV3).

P5 L2 What about "Anthropogenic bottom-up emission data" or something similar?

→ We changed "emissions" to "anthropogenic bottom-up emission inventories".

P5 L4: The bottom row of Figure 1 is for only toluene or TOL (toluene +less reactive aromatics)?

→ It is model emission data that represents the sum of toluene and less reactive aromatics. We agree that it would be confusing to the readers whether it is toluene or TOL (toluene + less aromatics). We added "(toluene + less reactive aromatics)" after "TOL" in the revised manuscript.

Please rearrange the order of figures and tables in the manuscript and supplementary document. Table S3 appears before Table S2. Similarly, Figure 7 needs to be rearranged.

→ Thanks for your comment. It is fixed now and Figure 7 is rearranged to Figure 2.

P6 L23: Only toluene or TOL emissions?

→ It is sum of toluene and less reactive aromatics. We changed "toluene" to "TOL (toluene + less reactive aromatics)" in the revised manuscript.

P7 L1-4: Is CO a major precursor affecting ozone formation?

→ CO is one of the important precursors but has less impact than $NO_2$ and VOC in urban area. The term "major" can lead to misunderstanding, suggesting that CO is the primary precursor significantly affecting $O_3$ concentrations. To avoid this confusion, we removed the term "major" in this line.

P7 L4: It would be great to include boxes of three regions in Figure 1.

→ Agreed. We changed Figure 1 as **Figure R4** in the revised manuscript.

[Figure]

**Figure R4**. The averaged spatial distribution map of the NO, CO, and TOL (toluene + less reactive aromatics) emissions from (a, d, g) EDGAR-HTAP v2, (b, e, h) v3, and (c, f, i) KORUS v5 in May.

P7 L12-18: What about other VOC emissions? Toluene is one of reactive species with OH, but isoprene, ethene, and other species are also reactive with OH, leading to ozone production. It would be helpful to show how different total reactive VOC emissions are between three inventories.

→ We added an additional row of total NMVOC amounts for each region in **Table S3** (Supporting information). The EDGAR-HTAP v3 has larger total non-methane VOC (NMVOC) emissions over China compared to EDATA-HTAP v2 and KORUS v5 by 38 and 27 % respectively. The descriptions of total NMVOC differences are added to the revised manuscript.

P7 L23-P8 L1: The sentence could potentially mislead about VOC emissions, as formaldehyde is produced by the oxidation of many VOCs. Some VOC species might overestimate or underestimate.

→ We changed "CO and VOC for all emissions by -40% (± 2%) and -25% (± 1%) (HCHO)" to ",HCHO, TOL, and XYL" to "CO, HCHO, TOL, and XYL for all emissions by -40% (± 2%), -25% (± 1%), -67% (± 21%), -53% (± 18%) respectively" to avoid misreading.

P8 L14-21: Regarding general comment #1, the authors should clarify the scope of this study for ozone underestimations.

→ The introduction and abstract sections are revised in the updated manuscript. It is changed as mentioned in the previous reply to general comment #1.

P9 L11-P10 L2: The paragraph could be separated at L19, and the sentence at L22-23 might be relocated after the description of $NO_2$ in South Korea. Please revise.

→ It is revised. And the L22-23 is relocated to P9 L17.

Section3.2:

It would be helpful to explain what causes discrepancies between models and observations. In addition, the authors mentioned that low MDA8 in the models could be related to low VOC emissions. However, determining the regimes to which certain areas belong should involve considering both NOx and VOCs. Underestimated ozone could result from both high NOx concentrations in the model and low VOC emissions. In Section 3.2, the comparison of ozone and $NO_2$ between models and observations was discussed separately. Combining the analysis and discussion would enhance the manuscript and the understanding of emissions.

→ We acknowledge the comment to the section 3.2. We anticipate that the reply to general comment #2 is enough to explain this reply as previously discussed about NOx and VOC emissions with FNR in **Table R1**.

P13 L16-19: The authors mentioned ground-based $NO_2$ observations over China and South Korea have positive biases in Section 2.3.2. If these positive biases are corrected, FNR might exceed 1, indicating a transition or NOx-limited regime. In addition, it would be useful to provide the range of the FNR to determine ozone production regimes.

→ As previously explained in the reply of general comment #2, we also analyzed FNR from different emission inventories for each regional box. The model FNR is not affected by molybdenum issue because it is the model value itself. In NCP and KOR (or SMA), all emission inventories show very low FNR value (< 1) indicating those areas as highly NOx-saturated regime. So, even though the $NO_2$ observations from molybdenum converter are corrected, the regime will not be dramatically changed.

Figure S5: Please plot $NO_2$ figures together with FNR and HCHO.

→ **Figure S5** is replaced to **Figure R6** in the revised Supporting Information.

[Figure]

**Figure R5**. Simulated surface (a-c) NO$_2$ and (d-f) HCHO concentrations and (g-i) HCHO to NO2 ratio (FNR) with (a, d, g) EDV2, (b, e, h) EDV3, and (c, f, i) KOV5 emissions for 14-16 LST. FNR greater than 1 is marked with black circles. The simulated NO$_2$, HCHO, and FNR are linearly interpolated to ground-based observation sites.

P14 L11-13: In Figure 2, ozone mixing ratios simulated with EDV3, KOV5 are not substantially lower than the observations, and MDA8 ozone also appears as yellow in Figure 3.

→ We wanted to mention that there are small biases in YRD region. To avoid misreading, we changed "The lower bias of O$_3$ in YRD" to "The reason why O$_3$ is well simulated in YRD, even though NO$_2$ is highly overestimated in this region," in the revised manuscript.

P14 L16-17: The lowest bias of -5.5 ppbv was observed for KOV5 in South Korea.

→ We wanted to emphasize that EDV3 shows the lowest absolute value of biases. For the $NO_2$, EDV3 shows the smallest bias in South Korea by -1.9 ppb. We changed 'the lowest' to 'the smallest' in the revised manuscript.

P16 L10: It would be helpful to describe why the authors showed HCHO.

→ We used HCHO as one of the proxies of VOC concentrations, though it is the product of many other VOCs in reaction with reactive gases. The sentence, "we also evaluated the model HCHO, which can be formed by oxidation of other VOCs but also directly emitted by anthropogenic sources, to investigate potential issue of anthropogenic VOC emissions", is added after "Additionally,".

P17 L5-6: Overall ozone production regimes were discussed in Section 3.2. However, to lead to your conclusion that underestimated ozone results from low VOCs during KORUS-AQ, it would be helpful to show FNR during KORUS-AQ. Also, the lifetime of CO is too long to significantly contribute to ozone. How much does the CO oxidation contribute to ozone? Is it related to background ozone concentration, and then how much?

→ We anticipate that the newly added Table 3 will explain the FNR in South Korea and SMA. It represents South Korea and SMA as highly NOx-saturated regime. We included **Figure R6** to the revised Supporting Information to show the CO contribution to $O_3$ concentrations in SMA. The reduced bias of CO derived from doubled China anthropogenic CO emissions (-96 to -63 ppb) slightly increased $O_3$ by 1.4 ppb compared to DC-8 for all flight observations. The overall descriptions of this results will be added at the P16 L9 to explain the CO impact on $O_3$ concentrations in SMA.

[Figure]

**Figure R6**. Vertically averaged (a) O$_3$ and (b) CO from DC-8 (black), EDV2 (sky blue), EDV3 (blue), EDV3 with double CO emission in China (EDV3 Ch2CO) (blue dashed), and KOV5 (red) 3 in SMA under 2 km height above ground level. The 1/2 of standard deviations are represented with black whiskers in each 200m layer. The sample number is presented with magenta color on the right side of the plots.

P17 L14-16: During KORUS-AQ, ground-based NO$_2$ was measured from a photolytic converter following the description in P10 L6-8, correct? If data from both molybdenum and photolytic converters were available, why did you choose to use data from molybdenum converter for the comparison? Also, low NOx from the model is a result influenced by various factors, including emissions, chemistry, PBL, and others. It would be helpful to describe which factors are affected rather than just saying the comparison with observations.

→ NO$_2$ observations by photolytic converter is only available at the Olympic Park. So, we used NO$_2$ observed by molybdenum converter, which is measurement instrument of Airkorea, for the analysis in KOR and SMA domain.

There are still other possibilities of $NO_2$ underestimations; 1) the emission factor used in this study is from Los Angeles basin that might be not adequate to SMA, 2) the uncertainty of HOx and ROx radicals from other sources can affect the $NO_2$ concentrations. We included those possible uncertainties at P17 L17.

P17 L 20-22: Simulated TOL from KOV5 also shows significant differences from observations below 1 km except at the surface.

→ "below 1 km" is replaced by "at surface level and had the lowest bias of -0.9 and -0.1 ppb respectively under 2 km".

P18 L13-14: It would be great to explain why surface and vertical columns exhibit different diurnal patterns.

→ The diurnal patterns of surface $NO_2$ concentration are attributed to the diurnal cycle of PBL height. In the morning, $NO_2$ is concentrated near the surface layer due to under-developing mixed layer height. In the afternoon, as the PBL height grow, $NO_2$ is more mixed and distributed to vertically higher altitudes. On the other hand, vertical column $NO_2$ density is high in the afternoon because of consistent emission of NOx during the daytime. Those explanations are included in the revised manuscript.

P18 L15: VCD patterns differ between model and Pandora.

→ "The simulated and observed HCHO show similar diurnal variations" is deleted.

P19 L 2-4: How did diurnal profiles contribute to reducing biases?

→ Diurnal profiles did not reduce the negative biases directly. We wanted to point out that TOL and XYL are still underestimated compared to surface observations at Olympic Park as

DC-8 shows underestimation of TOL and XYL under 2 km in SMA. We revised "reduced the model negative biases from EDV2 and EDV3" to "exhibited smaller negative biases than EDV2 and EDV3" to avoid confusion.

P19 L18-19: Even though the authors cite Peterson et al. (2019) to separate local and transport cases, please provide a brief description for readers.

→ "Stagnant and Blocking is the period that large anticyclone is located over South Korea, and Transport case is the period that South Korea is largely affected by long-range transport of air pollutants by westerly wind" is added after the sentence.

P20 L22-24: When considering the contributions of South Korea's CO and VOCs emissions, calculated as the difference between ch2 and chko2, the increase of CO and VOC emissions does not lead to improvements in CO and VOCs. The difference in CO is only 5 ppbv, while the difference in HCHO is 32 ppb, resulting in overestimation of the model. In addition, in chko2, doubling emissions in China might affect $O_3$, CO, and HCHO in Korea because of transport. Also, local cases may be influenced by the transport of species with relatively long lifetimes. To describe the effects of South Korea's emissions on ozone, CO, and HCHO, the author could simulate the model with only doubling CO and VOC emissions in Korea.

→ To avoid the transport impact when interpreting the emissions in South Korea, we conducted an additional simulation using EDGAR-HTAP v3 with double CO and VOC emission in South Korea only (EDV3_Ko2). In the Local case, although the EDV3_Ko2 reduced biases of $O_3$, CO, and HCHO over the SMA, doubled CO and VOC emissions in both South Korea and China (EDV3_ChKo2) showed the lowest biases. For the Transport case, doubling CO and VOC emissions in South Korea (EDV3_Ko2) slightly reduced $O_3$ and CO biases, but resulted in an overprediction of HCHO. We included those results in section 3.4 in the revised manuscript.

[Figure]

**Figure R7**. The biases in (a) the model O$_3$, (b) CO, and (c) HCHO concentrations (bars) relative to the DC-8 observations under 2 km height over SMA (dark gray: EDV3, red: EDV3 Ch2, orange: EDV3 Ko2, red: EDV3_ChKo2): (left panel) Local and (right panel) Transport case. Fractional differences (%) are shown in the white boxes.

Figures 2-6: Adding boxes to the plots in the second and third rows would be helpful for easily recognizing the regions.

→ The boxes are included in Figure 3, 5, and 6.

TableS4: Please provide definitions for the species names in MOZART and SAPRC-99.

→ The Table R2-R3 are included in the revised Supporting Information.

**Table R2**. The list of MOZART species (Emmons et al., 2010).

| Species | Atomic composition | Note |
|---------|-------------------|------|
| ISOP | $C_5H_8$ | isoprene |
| SO2 | $SO_2$ | sulfur dioxide |
| NO | NO | nitric oxide |
| NO2 | $NO_2$ | nitrogen dioxide |
| CO | CO | carbon monoxide |
| C2H6 | $C_2H_6$ | ethane |
| C2H5OH | $C_2H_5OH$ | ethanol |
| CH3OH | $CH_3OH$ | methanol |
| C3H8 | $C_3H_8$ | propane |
| BIGALK | $C_5H_{12}$ | lumped alkanes C>3 |
| TOLUENE | $C_6H_5(CH_3)$ | lumped aromatics |
| C2H4 | $C_2H_2$ | ethene |
| BIGENE | $C_4H_8$ | lumped alkenes C>3 |
| CH2O | $CH_2O$ | formaldehyde |
| CH3CHO | $CH_3CHO$ | acetaldehyde |
| CH3COCH3 | $CH_3COCH_3$ | acetone |
| MEK | $CH_3C(O)CH_2CH_3$ | methyl ethyl ketone |
| NH3 | $NH_3$ | Ammonia |

**Table R3**. The list of SAPRC99 species (Carter, 2000).

| Species | Note |
|---|---|
| ISOP | Isoprene |
| SO2 | Sulfur dioxide |
| NO | Nitric oxide |
| NO2 | Nitrogen dioxide |
| CO | Carbon monoxide |
| ALK1 | Alkanes and other non-aromatic compounds that react only with OH, and have kOH $< 5 \times 10^2$ ppm-1 min-1. (Primarily ethane) |
| ALK2 | Alkanes and other non-aromatic compounds that react only with OH, and have kOH between $5 \times 10^2$ and $2.5 \times 10^3$ ppm$^{-1}$ min$^{-1}$. (Primarily propane and acetylene) |
| ALK3 | Alkanes and other non-aromatic compounds that react only with OH, and have kOH between $2.5 \times 10^3$ and $5 \times 10^3$ ppm$^{-1}$ min$^{-1}$. |
| ALK4 | Alkanes and other non-aromatic compounds that react only with OH, and have kOH between $5 \times 10^3$ and $1 \times 10^4$ ppm$^{-1}$ min$^{-1}$. |
| ALK5 | Alkanes and other non-aromatic compounds that react only with OH, and have kOH greater than $1 \times 10^4$ ppm$^{-1}$ min$^{-1}$ |
| ARO1 | Aromatics with kOH $< 2 \times 10^4$ ppm$^{-1}$ min$^{-1}$. |
| ARO2 | Aromatics with kOH $> 2 \times 10^4$ ppm$^{-1}$ min$^{-1}$. |
| MEOH | Methanol |
| ETHE | Ethene |
| OLE1 | Alkenes (other than ethene) with kOH $< 7 \times 10^4$ ppm$^{-1}$ min$^{-1}$. (Primarily terminal alkenes) |
| PHEN | Phenol |
| CRES | Cresols |
| HCHO | Formaldehyde |
| CCHO | Acetaldehyde and Glycolaldehyde |
| RCHO | Lumped C3+ Aldehydes |
| BALD | Aromatic aldehydes (e.g., benzaldehyde) |
| GLY | Glyoxal |
| MGLY | Methyl Glyoxal |
| BACL | Biacetyl |
| MACR | Methacrolein |
| ACET | Acetone |
| MEK | Ketones and other non-aldehyde oxygenated products which react with OH radicals slower than $5 \times 10^{-12}$ cm$^3$ molec$^{-2}$ sec$^{-1}$ |
| PRD2 | Ketones and other non-aldehyde oxygenated products which react with OH radicals faster than $5 \times 10^{-12}$ cm$^3$ molec$^{-2}$ sec$^{-1}$ |
| MVK | Methyl Vinyl Ketone |
| IPRD | Lumped isoprene product species |
| NH3 | Ammonia |

Table S7-S8: Please clarify the time period for the data.

→ Those are corrected in the revised manuscript. We added 'for the KORUS-AQ campaign period' in the Table captions.

Figure 8: Please indicate the color for observations.

→ It is corrected in the revised manuscript.

**References**

Carter, W. P.: Documentation of the SAPRC-99 chemical mechanism for VOC reactivity assessment, Contract, 92, 95–308, https://intra.engr.ucr.edu/~carter/pubs/s99doc.pdf (last access: 9 June 2023), 2000.

Choi, J., Henze, D. K., Cao, H., Nowlan, C. R., Abad, G. G., Kwon, H.-A., Lee, H.-M., Oak, Y. J., Park, R. J., Bates, K. H., Massakkers, J. D., Wisthaler, A., and Weinheimer, A. J.: An Inversion Framework for Optimizing Non-Methane VOC Emissions Using Remote Sensing and Airborne Observations in Northeast Asia During the KORUS-AQ Field Campaign, J. Geophys. Res. Atmos., 127, e2021JD035844, https://doi.org/10.1029/2021JD035844, 2022.

Emmons, L. K., Schwantes, R. H., Orlando, J. J., Tyndall, G., Kinnison, D., Lamarque, J.-F., Marsh, D., Mills, M. J., Tilmes, S., Bardeen, C., Buchholz, R. R., Conley, A., Gettelman, A., Garcia, R., Simpson, I., Blacke, D. R., Meinardi, S., and Pétron, G.: The Chemistry Mechanism in the Community Earth System Model version 2 (CESM2), J. Adv. Model. Earth Syst., 12, e2019MS001882, https://doi.org/10.1029/2019MS001882, 2020.

Kim, S.-W., Kim, K.-M., Jeong, Y., Seo, S., Park, Y., and Kim J.: Changed in surface ozone in South Korea on diurnal to decadal timescales for the period of 2001-2021, Atmos. Chem. Phys., 23, 12867-12886, https://doi.org/10.5194/acp-23-12867-2023, 2023.

Goldberg, D. L., Saide, P. E., Lamsal, L. N., de Foy, B., Lu, Z., Woo, J.-H., Kim, Y., Kim, J., Gao, M., Carmichael, G., and Streets, D. G.: A top-down assessment using OMI NO2 suggests an underestimate in the NOx emissions inventory in Seoul, South Korea, during KORUS-AQ, Atmos. Chem. Phys., 19, 1801-1818, https://doi.org/10.5194/acp-19-1801-2019, 2019.

Miyazaki, K., Sekiya, T., Fu, D., Bowman, K. W., Kulawik, S. S., Sudo, K., Walker, T., Kanaya, Y., Takigawa, M., Ogochi, K., Eskes, H., Boersma, K. F., Thompson, A. M., Gaubert, B., Barre, J., and Emmons, L. K.: Balance of Emission and Dynamical Controls on Ozone During the Korea-United States Air Quality Campaign From Multiconstituent Satellite Data Assimilation, J. Geophys. Res. Atmos., 124, 387-413, https://doi.org/10.1029/2018JD028912 , 2019.

Souri, A. H., Nowlan, C. R., Abad, G. G., Zhu, L., Blake, D. R., Fried, A., Weinheimer, A. J., Wisthaler, A., Woo, J.-H., Zhang, Q., Chan Miller, C. E., Liu, X., and Chance, K.: An inversion of NOx and non-methane volatile organic compound (NMVOC) emissions using satellite observations during the KORUS-AQ campaign and implications for surface ozone over East Asia, Atmos. Chem. Phys., 20, 9837-9854, https://doi.org/10.5194/acp-20-9837-2020, 2020.

Tang, W., Emmons, L. K., Arellano Jr, A. F., Gaubert, B., Knote, C., Tilmes, S., Buchholz, R. R., Pfister, G. G., Diskin, G. S., Blake, D. R., Blake, N. J., Meinardi, S., DiGangi, J. P., Choi, Y., Woo, J.-H., He, C., Schroeder, J. R., Suh, I., Lee, H.-J., Kanaya, Y., Jung, J., Lee, Y., and Kim, D.: Source Contributions to Carbon Monoxide Concentrations During KORUS-AQ Based on CAM-chem Model Applications, J. Geophys. Res. Atmos., 124, 2796-2822, https://doi.org/10.1029/2018JD029151, 2019.

Wiedinmyer, C., and Emmons, L.: Fire Inventory from NCAR version 2 Fire Emission, Research Data Archive at the National Center for Atmospheric Research, Computational

and Information Systems Laboratory, https://doi.org/10.5065/XNPA-AF09, 2022. last access: 17 Oct 2023.

---

## Author Response (AR2)

Response to referee 1's Comments

Thank you for your time and helpful comments. Reviewer's comments are written in gray and our responses are in black.

The authors responded thoroughly to my comments on their earlier draft and improved the paper. I think the addition of Section 4 (strategies for improving $O_3$ simulations) is a good one; can you include some discussion of Ox in this section as well? After addressing this, I recommend publication.

→ We added analysis of Ox with surface observations in China and South Korea in Section 4 and Supporting Information. The metrices and bar figures of Ox were added to Supporting Information (**Table S14-S17, Figure S16 and S17**). Please refer to Figure R1 and R2 for the analysis. We added the new paragraphs to the revised manuscript. Additionally, we have made slight adjustments to the numbers in Table S14-S17, in the range of 1-2%, to account for missing observations. We have made minor language edits to incorporate the opinions of co-authors.

For regions (Figure R1 or Figure S16):

The biases of $O_X$ typically follow $O_3$ biases across cases in all regions except NCP, YRD, PRD, and NOC, which experience high $NO_X$ conditions. Refer to Supporting Information Figure S16 for analysis of $O_X$ along with $O_3$ across various regions. In these specific regions, a substantial reduction in $NO_X$ levels (as in C4 and C7) resulted in an increase in $O_3$ bias, while there was a decrease in $O_X$.

For cities (Figure R2 or Figure S17)

The biases of $O_X$ generally follow $O_3$ biases in Chengdu and Chongqing, where the simulated $O_3$ initially exhibits a notably high positive bias (50-60%), attributable to high VOC. Refer to Supporting Information Figure S17 for an analysis of $O_x$ and $O_3$ across cases and cities. In contrast, for other cities experiencing high $NO_X$ conditions with positive $NO_2$ biases, a

reduction in $NO_X$ levels (as in C4 and C7) led to a decrease in $O_X$ (and its bias for most cities). However, there was a simultaneous increase in $O_3$ and its bias, attributed to the $NO_X$-saturated regime (Figure S17).

[Figure]

**Figure R1**. Same as **Figure 14** except that $NO_2$ is changed to $O_X$ (= $NO_2$ + $O_3$).

[Figure]

**Figure R2**. Same as **Figure 15** except that $NO_2$ is changed to Ox ($= NO_2 + O_3$).